# FROM PARAMETERS TO BEHAVIORS: UNSUPERVISED COMPRESSION OF THE POLICY SPACE

**Davide Tenedini ♣, Riccardo Zamboni ♣, Mirco Mutti ♠, Marcello Restelli ♣**

♣ Politecnico di Milano ♠ Technion

{davide.tenedini, riccardo.zamboni, marcello.restelli}@polimi.it
mirco.m@technion.ac.il

## ABSTRACT

Despite its recent successes, Deep Reinforcement Learning (DRL) is notoriously sample-inefficient. We argue that this inefficiency stems from the standard practice of optimizing policies directly in the high-dimensional and highly redundant parameter space $\Theta$. This challenge is greatly compounded in multi-task settings. In this work, we develop a novel, unsupervised approach that compresses the policy parameter space $\Theta$ into a low-dimensional latent space $\mathcal{Z}$. We train a generative model $g : \mathcal{Z} \to \Theta$ by optimizing a behavioral reconstruction loss, which ensures that the latent space is organized by functional similarity rather than proximity in parameterization. We conjecture that the inherent dimensionality of this manifold is a function of the environment's complexity, rather than the size of the policy network. We validate our approach in continuous control domains, showing that the parameterization of standard policy networks can be compressed up to five orders of magnitude while retaining most of its expressivity. As a byproduct, we show that the learned manifold enables task-specific adaptation via Policy Gradient operating in the latent space $\mathcal{Z}$.

## 1 INTRODUCTION

High-dimensional parameterization of policies via deep neural networks has been a key driver of recent successes in Deep Reinforcement Learning (among others, Andrychowicz et al., 2020; Smith et al., 2022; Bakhtin et al., 2022; Wurman et al., 2022; Degrave et al., 2022). A major drawback of this approach, however, is a significant increase in sample complexity, which is further compounded when the agent is called to solve multiple and potentially unknown tasks, typically requiring learning *tabula rasa* (Agarwal et al., 2022). This inefficiency often stems from a fundamental redundancy in the parameter space, where a large set of distinct weight configurations maps to a much smaller set of effective behaviors. Various approaches tried to solve this limitation as a *byproduct*, such as explicitly learning diverse behaviors (Eysenbach et al., 2018; Zahavy et al., 2023; De Paola et al., 2025; Zamboni et al., 2025), or enforcing small policy networks in asymmetric actor-critic architectures (Degrave et al., 2022; Mastikhina et al., 2025).

In this paper, we address this limitation directly through the lens of the *Manifold Hypothesis* (Cayton et al., 2005), a widely accepted tenet of Machine Learning, and we hypothesize that it holds in Reinforcement Learning as well, namely that:

> *The manifold of realizable behaviors is intrinsically low-dimensional and largely independent of the network's parameter count.*

In view of this hypothesis, we propose a paradigm shift from learning in the parameter space to learning in the (latent) *behavior* space itself. To do so, the agent first needs to learn a latent representation of the possible behaviors, which, according to the aforementioned hypothesis, should be *low-dimensional* and *policy network invariant*. Then, it needs to find a way to leverage this representation to solve different tasks *efficiently*. The proposed solution is a novel two-stage framework

directly inspired by the *Unsupervised* RL formalism (Laskin et al., 2021), allowing for the *explicit* exploitation of this latent structure. In a first *pre-training* phase, we learn a latent representation of the behavior manifold by leveraging a generative model in a fully *unsupervised* fashion, that is, without including any information related to a specific task, i.e., reward. In this way, we can learn a latent structure that models the intrinsic nature of the environment dynamics, rather than its coupling with a task, and preserve the end-to-end differentiability that makes gradient-based optimization effective. In a second *fine-tuning* phase, we leverage the pre-trained representation to fine-tune policies on specific tasks known *a posteriori*, avoiding the need to learn from scratch. In particular, the fine-tuning phase involves performing gradient steps in the latent space, thereby optimizing *latent behaviors* directly. This approach enables the agent to explore the inherently low-dimensional behavior space rather than the high-dimensional parameter space.

In this paper, we address the following:

> **Research Questions:**
>
> (**Q1**) Is it possible to learn a low-dimensional latent representation of a high-dimensional policy parameter space in an unsupervised fashion?
>
> (**Q2**) What are the properties of such a latent representation, if it exists? Is its intrinsic dimension a function of the behavioral complexity rather than the size of the parameter space?
>
> (**Q3**) How can we fine-tune on specific tasks leveraging the low-dimensional space? Does this offer any advantages?

**Content Outline and Contributions.** First, in Section 3, we formulate the problem of learning a latent representation of behaviors in an unsupervised fashion and then leveraging it to solve specific tasks. Then, in Section 4, we characterize our proposed solution to this problem, namely, addressing it in a two-stage pipeline. Finally, in Section 5, we perform experiments extensively to address the Research Questions. We demonstrate that the proposed pipeline is indeed able to learn low-dimensional latent representations (**Q1**), which are more influenced by the environment than by the size of the compressed policies (**Q2**). Finally, we demonstrate that learning over this reduced space can make simple algorithms competitive against complex state-of-the-art DRL algorithms (**Q3**).

## 2 PRELIMINARIES

**Notation.** In the following, we denote a set with a calligraphic letter $\mathcal{A}$ and its size as $|\mathcal{A}|$. For a finite set, $\Delta(\mathcal{A})$ denotes the probability simplex. For continuous spaces, we denote the set of Borel probability measures as $\mathcal{P}(\mathcal{A})$. When the context allows, we use $\Delta(\mathcal{A})$ to refer generally to the space of valid distributions over $\mathcal{A}$. For two distributions $p_1, p_2 \in \Delta(\mathcal{A})$, we define a general measure of divergence between distributions with $D(p_1 \parallel p_2)$.

**Interaction Protocol.** As a base model for interaction, we consider a (finite-horizon) Controlled Markov Process (CMP). A CMP is defined as the tuple $\mathcal{M} := (\mathcal{S}, \mathcal{A}, \mathbb{P}, \mu, T)$, where $\mathcal{S}$ is the state space and $\mathcal{A}$ is the action space. At the start of an episode, the initial state $s_0$ of $\mathcal{M}$ is drawn from an initial state distribution $\mu \in \Delta(\mathcal{S})$. Upon observing $s_0$, the agent takes action $a_0 \in \mathcal{A}$, and the system transitions to $s_1 \sim \mathbb{P}(\cdot \mid s_0, a_0)$ according to the transition model $\mathbb{P} : \mathcal{S} \times \mathcal{A} \to \Delta(\mathcal{S})$. The process is repeated until $T$ is reached and $s_T$ is generated, with $T < \infty$ being the horizon of an episode. The agent selects actions according to a decision *policy* $\pi : \mathcal{S} \to \Delta(\mathcal{A})$ such that $\pi(a \mid s)$ denotes the conditional probability of taking action $a$ upon observing state $s$. Deploying a policy $\pi$ over $\mathcal{M}$ leads to the generation of trajectories $\tau$, defined as a sequence of state-action pairs $\tau := (s_0, a_0, s_1, a_1, \ldots, s_T)$. Furthermore, a policy $\pi$ induces a state distribution $d_\pi^s \in \Delta(\mathcal{S})$ over the state space of the CMP $\mathcal{M}$ defined as $d_\pi^s(s) = \frac{1}{T+1} \sum_{t=0}^{T} \Pr(s_t = s)$. It also induces a state-action distribution $d_\pi^{sa} \in \Delta(\mathcal{S} \times \mathcal{A})$, defined as $d_\pi^{sa}(s, a) = \pi(a \mid s) \, d_\pi^s(s)$, which we will denote as the *behavior* of the policy. In the following, we will consider deterministic policies $\pi_\theta : \mathcal{S} \to \mathcal{A}$ represented by neural networks parameterized by a set of weights $\theta \in \Theta$, where $\Theta \subseteq \mathbb{R}^P$ is the policy parameter space, with $P$ being the total number of parameters. We define the *Policy Space* $\Pi_\Theta$ as the collection of policies that can be represented by $\Theta$. For brevity of notation, we denote a policy $\pi_\theta$ as its set of parameters $\theta$ and the policy space $\Pi_\Theta$ as the parameter space $\Theta$ that induces it.

**(Unsupervised) RL.** In RL, an agent learns how to solve (downstream) *tasks*, encoded by different reward signals. To this end, we define a Markov Decision Process (MDP, Puterman, 2014) $\mathcal{M}_R := \mathcal{M} \cup R$ as a coupling of a CMP $\mathcal{M}$ and a reward function $R : \mathcal{S} \times \mathcal{A} \to \mathbb{R}$, which the agent observes after every state transition. In the **Unsupervised Reinforcement Learning** (URL, Laskin et al., 2021) framework, the reward signal is not always available to the agent from the beginning. It often belongs to a (potentially infinite) family of tasks $\mathcal{R}$, also unknown to the agent. URL is then composed of two phases: **(1)** an *unsupervised pre-training* phase involves the agent interacting with a CMP to acquire general-purpose knowledge without receiving any reward signal, which is distilled into a pre-trained model $\mathbb{M}$; **(2)** the *supervised fine-tuning* phase begins once a reward function $R \in \mathcal{R}$ is revealed. At this point, the CMP becomes a standard MDP $\mathcal{M}_R$, and the agent leverages the pre-trained model $\mathbb{M}$ to find a set of policy parameters that maximizes the expected return for the given task, namely as:

$$\boldsymbol{\theta}^* = \arg\max_{\boldsymbol{\theta} \in \Theta} J^R(\boldsymbol{\theta}, \mathbb{M}) = \arg\max_{\boldsymbol{\theta} \in \Theta} \mathbb{E}_{(s,a) \sim d^{sa}_{\pi_{\boldsymbol{\theta}}}, \mathbb{M}} \left[ R(s, a) \right]. \tag{1}$$

**Policy Optimization.** Policy Optimization (PO, Deisenroth et al., 2013), which involves optimizing the policy parameters directly, has shown surprisingly good results. This is especially true for deep neural policies, where first-order methods have been extensively employed. A popular approach to PO is **Policy Gradient** (PG, Peters & Schaal, 2008), which updates the parameters by simple gradient ascent $\boldsymbol{\theta}' \leftarrow \boldsymbol{\theta} + \alpha \nabla_{\boldsymbol{\theta}} J(\boldsymbol{\theta})$. Among others, **Policy Gradient with Parameter-based Exploration** (PGPE, Sehnke et al., 2008) is a PG algorithm that handles exploration in the parameter space by sampling the policy parameters $\boldsymbol{\theta}$ from a hyper-policy $\nu_{\boldsymbol{\phi}}$, parameterized by $\boldsymbol{\phi}$.[1] PGPE optimizes a trajectory-based version of the objective defined in Eq. 1, defined as:

$$J^R(\boldsymbol{\theta}, \boldsymbol{\phi}, \mathbb{M}) = \mathbb{E}_{\tau \sim p(\cdot \mid \boldsymbol{\theta}), \boldsymbol{\theta} \sim \nu_{\boldsymbol{\phi}}, \mathbb{M}}[R(\tau)], \tag{2}$$

where $R(\tau) = \sum_{t=0}^{T} R(s_t, a_t)$ is the return of a trajectory, and $p(\tau \mid \boldsymbol{\theta}) = \mu(s_0) \prod_{t=0}^{T} \mathbb{P}(s_{t+1} \mid s_t, a_t) \pi_{\boldsymbol{\theta}}(a_t \mid s_t)$ is the probability density of a trajectory. In PGPE, the parameter vector $\boldsymbol{\phi}$ is usually updated via gradient ascent using a Monte Carlo estimator of the gradient computed over $N \in \mathbb{N}$ trajectories:

$$\hat{\nabla}_{\boldsymbol{\phi}} J^R(\boldsymbol{\theta}, \boldsymbol{\phi}) = \frac{1}{N} \sum_{i=1}^{N} \nabla_{\boldsymbol{\phi}} \log \nu_{\boldsymbol{\phi}}(\boldsymbol{\theta}_i) R(\tau_i). \tag{3}$$

**Generative Models.** Generative models have achieved remarkable success in density estimation for multi-modal data, drawing significant interest from the RL community. Among others, **Autoencoders** (AE, Hinton & Salakhutdinov, 2006) are a type of artificial neural network used to learn efficient data encoding in an unsupervised manner. The aim is first to learn encoded representations of data and then generate the input data (as closely as possible) from the learned encoded representations. More specifically, their goal is to map a *data space* $\mathcal{X} \subseteq \mathbb{R}^n$ to a *latent space* $\mathcal{Z} \subseteq \mathbb{R}^k$, with $k \ll n$. AEs are composed of an encoder, a function $f_{\boldsymbol{\xi}} : \mathcal{X} \to \mathcal{Z}$, parameterized by vector $\boldsymbol{\xi}$, which maps a data sample $\boldsymbol{x} \in \mathcal{X}$ to a latent code $\boldsymbol{z} \in \mathcal{Z}$, and a decoder, a function $g_{\boldsymbol{\zeta}} : \mathcal{Z} \to \mathcal{X}$, parameterized by vector $\boldsymbol{\zeta}$, which reconstructs the data sample $\hat{\boldsymbol{x}} \in \mathcal{X}$ from the latent code $\boldsymbol{z}$ in such a way that $g_{\boldsymbol{\zeta}} \approx f_{\boldsymbol{\xi}}^{-1}$. An AE is typically trained by minimizing the reconstruction error $\mathcal{L}_{AE}(\boldsymbol{x}) = d(\boldsymbol{x}, (g_{\boldsymbol{\zeta}} \circ f_{\boldsymbol{\xi}})(\boldsymbol{x}))$, where $d$ is a metric that measures the distance of samples in the data space. These architectures are particularly compelling in light of the Manifold Hypothesis (Cayton et al., 2005): AEs learn the underlying structure by compressing the data into a compact latent space that represents the manifold and then reconstructing the original data from it, as illustrated in Fig. 1.

Unfortunately, AEs are far from being bullet-proof. In cases where no plausible embedding exists, even networks $(f_{\boldsymbol{\xi}}, g_{\boldsymbol{\zeta}})$ which come close to perfectly reconstructing the manifold $\mathcal{M}$ will incur numerical instability (Cornish et al., 2020). In some other cases, it is possible to resolve these topological issues by increasing the latent dimension $k$. For instance, a dimensionality of $k = 2d^\star + 1$ is enough to topologically embed any manifold of dimension $d^\star$ in $\mathbb{R}^k$ (Theorem V3, Hurewicz & Wallman, 2015).

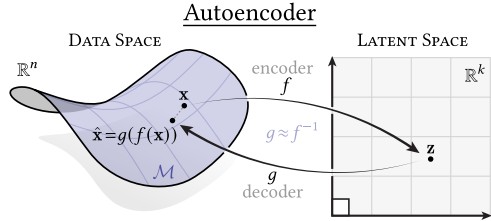

Figure 1: Autoencoder Spaces and Data Manifold.

---

[1]For instance, Gaussian hyper-policies will be parameterized by their mean and standard deviation, i.e. $\phi = (\boldsymbol{\mu}, \boldsymbol{\sigma})$.

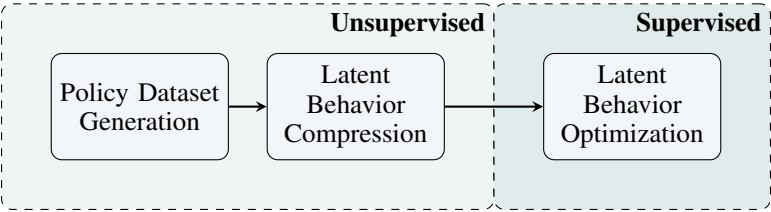

Figure 2: Pipeline of Unsupervised Compression of the Policy Space.

## 3 PROBLEM FORMULATION

By looking closely to Eq. 1, one should notice that to solve an RL task, the agent *just* needs to focus on visiting the states and actions that *matter for the task*. Yet, this simple intuition hides a few traps. First of all, different policy parameters $\boldsymbol{\theta} \in \Theta$ might induce nearly identical distributions over actions. Yet, even different distributions over actions could lead to comparable state-action distributions due to the complex structure of the environment. Finally, in almost all problems of interest, there may be multiple and potentially unknown tasks that the agent could be called upon to solve, and it would be risky to deem any state-action distribution irrelevant without additional information on the task structure.

In this work, we aim to address these issues by focusing on *behaviors* rather than parameters, under the lens of the Manifold Hypothesis: we want to learn a latent manifold of realizable behaviors, and we do this by *compressing* parameters inducing similar behaviors to the same latent representation. For a policy parameters space $\Theta \subseteq \mathbb{R}^n$, we define $\mathcal{Z} \subseteq \mathbb{R}^k$ as a $k$-dimensional latent space, with $k \ll n$, and we look for a function $g : \mathcal{Z} \rightarrow \Theta$ that maps a latent vector $\boldsymbol{z} \in \mathcal{Z}$, which we also refer to as *latent code*, to a corresponding policy parameter vector $\boldsymbol{\theta} = g(\boldsymbol{z})$. As a result, any policy could be written as $\pi_{\boldsymbol{\theta}} = \pi_{\boldsymbol{\theta}=g(\boldsymbol{z})} = \pi_{\boldsymbol{z}}$.

We refer to this problem as **Latent Behavior Compression**, which is formally defined as finding the generative function $g^{\star} : \mathcal{Z} \rightarrow \Theta$, such that:

$$\forall \boldsymbol{\theta} \in \Theta, \quad \exists \boldsymbol{z} \in \mathcal{Z} : \quad g^{\star} = \arg\min_{g} D\left(d_{\pi_{\boldsymbol{\theta}}}^{sa} \parallel d_{\pi_{\boldsymbol{z}}}^{sa}\right). \tag{4}$$

This task is essentially *unsupervised*, as any notion of a specific task is absent. Indeed, it is somewhat similar to the Policy Space Compression framework (Mutti et al., 2022), yet in the latter, the authors aim to reduce the cardinality of the policy space, rather than its dimensionality. Moreover, the constraints defining a valid compression are stricter than ours, resulting in an optimization problem that is known to be NP-hard.

Once such a low-dimensional space and generative function are available, solving for different tasks will require searching over a simpler space than the original one. We call this process **Latent Behavior Optimization**. In other words, the standard PO problem of Eq. 1, which requires finding an optimal policy parameter vector $\boldsymbol{\theta}^* \in \Theta$, will be reformulated as the problem of finding an optimal latent code $\boldsymbol{z}^* \in \mathcal{Z}$ that, via the generative function $g$, yields $\boldsymbol{\theta}^*$. For a given task with reward $R \in \mathcal{R}$, the policy optimization problem is now defined as:

$$\boldsymbol{z}^* = \arg\max_{\boldsymbol{z} \in \mathcal{Z}} J^R(\boldsymbol{z}) = \arg\max_{\boldsymbol{z} \in \mathcal{Z}} J^R\left(\boldsymbol{\theta} = g\left(\boldsymbol{z}\right)\right).$$

Contrary to the Latent Behavior Compression task, this task is essentially *supervised*, as it is well-defined as soon as the agent is provided with a reward. In the following, we will show how the URL framework can indeed provide essential tools in addressing the two problems.

## 4 METHOD: UNSUPERVISED COMPRESSION OF THE POLICY SPACE

To address the sample inefficiency inherent in high-dimensional policy parameter spaces, we propose a paradigm shift from directly optimizing in the parameter space to learning within a compact, low-dimensional policy manifold that captures the true diversity of behaviors. This is achieved through a two-phase framework: a completely unsupervised, task-agnostic pre-training phase to discover the manifold, followed by a supervised, task-specific fine-tuning phase. As illustrated in Fig. 2, this framework is composed of three steps: (**1**) generating a behaviorally diverse dataset of policies, (**2**) learning the latent policy manifold via a generative model, and (**3**) performing fine-tuning by optimizing over this learned latent space.

**Policy Dataset Generation.** Many manifold reconstruction algorithms depend on efficiently covering the manifold with samples (Bernstein et al., 2000; Cheng et al., 2005; Fefferman et al., 2016). Thus, the first stage of our framework involves generating a dataset $\mathcal{D}_\Theta$ of policies intended to cover the manifold of behaviorally diverse policies.

A naïve option is to randomly sample $N$ policies, $\hat{\mathcal{D}}_\Theta = \{\boldsymbol{\theta}_i\}_{i=1}^N$, by drawing their parameters from a uniform distribution. Unfortunately, it is well-understood that such naïve sampling from the parameter space is unlikely to produce uniform coverage of the behavior space, as it tends to favor functionally similar, often non-exploring, policies.

To address this bias, an explicit measure of *behavioral diversity* is needed. Looking at Eq. 4, one notices that optimizing such a measure directly requires estimating the divergence between two state-action distributions $d^{sa}_{\pi_{\boldsymbol{\theta}}}, d^{sa}_{\pi_{\boldsymbol{\theta}'}}$. Unfortunately, this would not only be computationally intensive but also require sampling from the environment for a potentially vast set of policies. To avoid this, we will take into account an upper bound to this quantity in the case of finite-horizon tasks (Prop. E.1, Metelli et al., 2018), namely $D(\pi_{\boldsymbol{\theta}} \parallel \pi_{\boldsymbol{\theta}'})$. In practice, we substitute this measure with the L2 distance of two policies in the action space, evaluated on a finite subset of the state space, or formally:

$$D(\pi_{\boldsymbol{\theta}} \parallel \pi_{\boldsymbol{\theta}'}) = \sqrt{\sum_{i=1}^M \left(\pi_{\boldsymbol{\theta}}\left(\cdot \mid s_i\right) - \pi_{\boldsymbol{\theta}'}\left(\cdot \mid s_i\right)\right)^2}.$$

Based on this proxy, we apply a Novelty Search algorithm (Lehman & Stanley, 2011) by computing a *novelty score*, $\rho(\pi_{\boldsymbol{\theta}})$, for each policy based on its average divergence from its $k_n$-nearest neighbors: $\rho(\pi_{\boldsymbol{\theta}}) = \frac{1}{k_n} \sum_{i=1}^{k_n} D(\pi_{\boldsymbol{\theta}} \parallel \pi_{\boldsymbol{\theta}_i})$.

Then, a high score indicates a behaviorally unique policy. Using this metric, we form the final dataset $\mathcal{D}_\Theta$ by selecting only the top percentile of policies with the highest novelty scores, ensuring a dataset of behaviorally diverse policies.

**Latent Behavior Compression.** In the second stage, we learn the low-dimensional manifold from the filtered policy dataset $\mathcal{D}_\Theta$. Potentially, any generative model would do the work. Still, here we are interested in learning latent low-dimensional representations while preserving the end-to-end differentiability that makes gradient-based optimization effective. For these reasons, we employ a symmetric autoencoder architecture with an encoder $f_{\boldsymbol{\xi}} : \Theta \rightarrow \mathcal{Z}$ and a decoder $g_{\boldsymbol{\zeta}} : \mathcal{Z} \rightarrow \Theta$. While a standard autoencoder minimizes a parameter reconstruction error, our goal is to compress policy *behavior*. We therefore introduce a novel **Behavioral Reconstruction Loss**, which trains the autoencoder to minimize the expected behavioral divergence between the original policy and its reconstruction:

$$\mathcal{L}_B(\boldsymbol{\xi}, \boldsymbol{\zeta}) = \mathbb{E}_{\boldsymbol{\theta} \sim \mathcal{D}_\Theta} \left[ D\left(\pi_{\boldsymbol{\theta}} \parallel \pi_{(g_{\boldsymbol{\zeta}} \circ f_{\boldsymbol{\xi}})(\boldsymbol{\theta})}\right) \right].$$

This objective frees the decoder from reproducing the exact parameter values, allowing it to discover any parameterization that generates the desired behavior. As a result, the latent space $\mathcal{Z}$ becomes organized purely by functional similarity, effectively capturing the policy manifold. In practice, we use an empirical estimator of the behavioral reconstruction loss based on the notion of divergence in the action space. For this purpose, we train our autoencoders to minimize the *Mean Squared Error* between action vectors relative to a subset of the state space sampled at each gradient step, resulting in the estimator $\hat{\mathcal{L}}_B(\boldsymbol{\xi}, \boldsymbol{\zeta}) = \frac{1}{NM} \sum_{i=1}^N \sum_{j=1}^M \left\| \pi_{\boldsymbol{\theta}_i}(s_j) - \pi_{(g_{\boldsymbol{\zeta}} \circ f_{\boldsymbol{\xi}})(\boldsymbol{\theta}_i)}(s_j) \right\|_2^2$, where $N$ is the number of policies, and $M$ is the number of sampled states.

**Latent Behavior Optimization.** In the final stage, we leverage the learned latent manifold for rapid, task-specific fine-tuning. With the decoder parameters $\zeta^\star$ frozen, $g_{\zeta^\star}$ becomes a deterministic and differentiable function that generates policies from latent codes. This structure allows us to adapt a wide range of PG methods to operate in the latent space. By applying the chain rule, the standard policy gradient can be back-propagated through the frozen decoder to update the latent code $z$:

$$\nabla_z J^R(z) = \nabla_z g_{\zeta^\star}(z)^\top \nabla_\theta J^R(\theta), \tag{5}$$

where $\nabla_\theta J^R(\theta)$ is the conventional policy gradient and $\nabla_z g_{\zeta^\star}(z)$ is the Jacobian of the decoder. This provides a general recipe for adapting popular PG algorithms to our framework. This approach is particularly advantageous for parameter-exploring PG methods, like PGPE, which notoriously struggle with high-dimensional parameter spaces. By operating on the low-dimensional latent space, these algorithms regain their effectiveness while still controlling the expressive power of the original large network.[2]

**Remarks.** In this section, we proposed three specific instantiations for each phase. Yet, we emphasize that the proposed pipeline represents the most relevant contribution *per se*, independently of how it is realized, i.e., how the policies are collected, which divergence measure is used, which generative model or PO algorithm over the latent space is employed.

## 5 EXPERIMENTS

We now investigate through extensive empirical corroboration how the proposed method addresses the research questions. In order to do so, we will mainly focus on the *unsupervised pre-training* phase of the proposed pipeline seen in Fig. 2, in which a latent representation is built out of general datasets of policies *not designed to address any specific task explicitly*, and we report the empirical results in Subsec. 5.1. Finally, we make sure that such a latent space can indeed be leveraged in later *supervised fine-tuning* phases *as soon as a task is provided*, and report the results in Subsec. 5.2. A detailed description of the environments and experimental settings can be found in Appendix B.

**Experimental Domains.** The experiments are performed to illustrate essential features of Latent Behavior Compression, and for this reason, the domains are selected for being challenging while keeping high interpretability. The first is the **Mountain Car Continuous** (MC, Moore, 1990) environment. To evaluate the quality and characteristics of the latent space, we define four downstream tasks: `standard` and `left` have the goal state on the right and left hill, respectively; `speed` and `height` incentivize the car to keep a high speed and vertical coordinate, respectively, without terminating the episode. We also consider three environments from the MuJoCo suite (Todorov et al., 2012; Tassa et al., 2018). For **Reacher** (RC), we define four downstream tasks: `speed`, which incentivizes the fingertip to move with high linear velocity; `clockwise` and `c-clockwise` reward the agent for each step the fingertip is rotating clockwise and counterclockwise, respectively; and `radial`, which promotes the retraction and extension of the arm. For **Hopper** (HP), we define four downstream tasks: `forward`, `backward`, and `standstill` reward the agent for positive, negative, or close-to-zero velocity along the $x$-axis respectively; `jump` rewards the agent for achieving a certain position along the $z$-axis. Finally, for **HalfCheetah** (HC), we define four downstream tasks: `forward` and `backward` are defined as for HP; `frontflip` and `backflip` reward the agent each time it performs a frontflip and backflip, respectively.

**Experimental Regimes.** The experiments are performed over a set of different parameters. In MC, we took into account three Policy Sizes (Small, Medium, and Large) with roughly $10^1, 10^3$, and $10^5$ parameters respectively, three Policy Dataset Sizes (10k, 50k, and 100k generated policies, with a 10% novelty-based cut-down), and three Latent Space Sizes (1D, 2D, and 3D). In RC, we focused on a specific configuration with Medium policies, Policy Dataset size of 100k, but five possible latent space sizes (1D, 2D, 3D, 5D, and 8D). In HP and HC, we focused on Policy Datasets of 10k, using Medium policies, with three latent space sizes (5D, 8D, and 16D).[3]

---

[2]Additionally, running PGPE over the latent space does not actually require computing the Jacobian of the decoder $\nabla_z g_{\zeta^\star}(z)$, as explained in Appendix B.

[3]Notably, the AE architecture and training hyper-parameters are left the same for every experiment, regardless of configuration or environment.

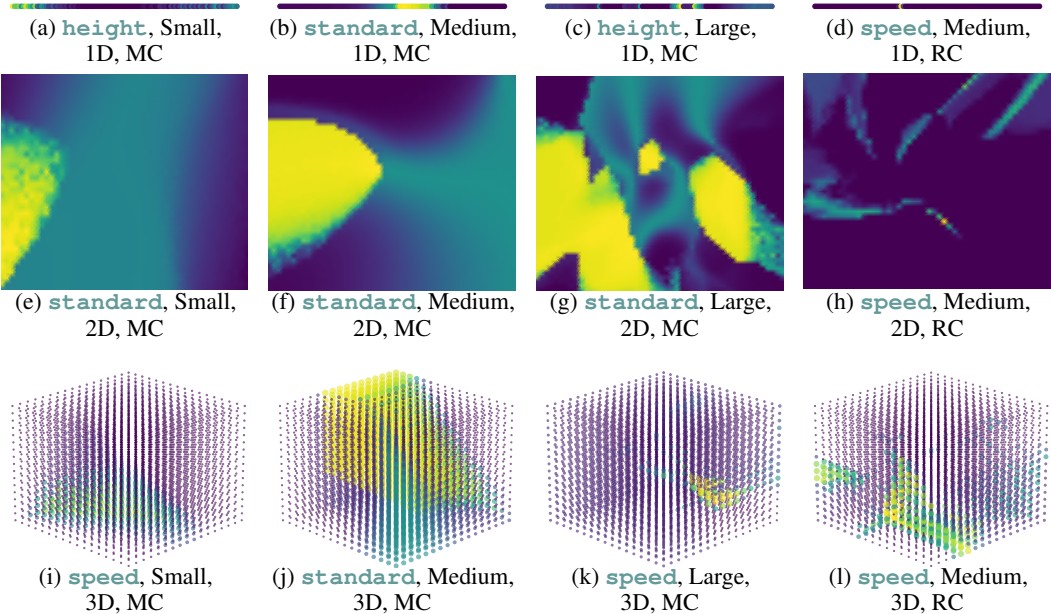

Figure 3: Landscape of the Latent Behavior Manifold. **Lighter** and **darker** colors indicate higher and lower returns of the decoded policy. The plots shown here represent a subset of the full results reported in Appendix B. We consider a specific seed with different tasks (`height`, `standard`, `speed`), policy size (Small, Medium, Large), and encoding dimension (1D, 2D, 3D), for both MC (first three columns, datasets of 50k policies) and RC (last column, datasets of 100k policies).

## 5.1 UNSUPERVISED PRE-TRAINING

First, we address the first two research questions, that is:

> (**Q1**) Is it possible to learn a low-dimensional latent representation of a high-dimensional policy parameter space in an unsupervised fashion?
>
> (**Q2**) What are the properties of such a latent representation, if it exists? Is its intrinsic dimension a function of the behavioral complexity rather than the size of the parameter space?

To do so, we discretize the latent space into a subset $\{z_i\}_{i=1}^N$ and perform evaluations of the decoded policies $\{\pi_{z_i}\}_{i=1}^N$. This allows for a rough estimate of the quality of the policies compressed in the latent manifold. Further details on how such discretization was performed are in Appendix B.

**Landscape of the Latent Behavior Manifold.** We visually inspected the latent spaces trained under different conditions and environments, and we report a handful in Fig. 3. Interestingly, it is apparent that the latent spaces, regardless of the choice of encoding dimension (top-to-bottom) or policy size (left-to-right), can encode *some* behaviorally diverse policies with high performance. For instance, in Fig. 3e,3f, and 3g, a 2D latent space can encode policies of all three sizes, but the landscape grows more complex with the larger policy sizes. We speculate that this is due to the increased range of behaviors expressed by larger policies and the hardness of high-compression regimes. Indeed, the same trend is present for different tasks, as in Fig. 3i,3k. On the other hand, by changing the encoding dimension as in Fig. 3b,3f,3j, it is clear how certain behavioral areas at high performance are able to grow larger, creating a better optimization landscape. Unfortunately yet, the compression is only as good as the dataset used to learn the latent space: when a behavior is scarcely represented in the dataset, as is the case for the task `height` in Fig. 3a,3c, it is unlikely that the learned representation will encode it in large areas, or encode it at all. As for RC, the compression architecture struggles to compress the policies at higher compression regimes (Fig. 3d,3h), as the environment is more challenging and presents a wider range of behaviors. On the other hand, large areas of good quality compression are present for larger dimensions of the encoding (Fig. 3l), confirming the expected theoretical behavior (Hurewicz & Wallman, 2015).

Table 1: Quality of Latent Behavior Compression in MC. We report the performance recovery for three tasks. We report mean and standard deviation computed over 3 seeds. **Bold** indicates best across latent dimensions for each experiment setting; shading indicates best per task.

| Config. | | Standard | | | Speed | | | Height | | |
|---|---|---|---|---|---|---|---|---|---|---|
| Policy | Dataset | 1D | 2D | 3D | 1D | 2D | 3D | 1D | 2D | 3D |
| Small | 10k | $0.51_{\pm.00}$ | $0.66_{\pm.11}$ | $\mathbf{0.74}_{\pm.16}$ | $0.15_{\pm.05}$ | $0.15_{\pm.08}$ | $\mathbf{0.27}_{\pm.06}$ | $0.16_{\pm.10}$ | $0.16_{\pm.09}$ | $\mathbf{0.27}_{\pm.05}$ |
| Small | 50k | $0.64_{\pm.19}$ | $0.93_{\pm.10}$ | $\mathbf{0.94}_{\pm.06}$ | $0.10_{\pm.10}$ | $0.42_{\pm.15}$ | $\mathbf{0.44}_{\pm.20}$ | $0.11_{\pm.11}$ | $0.45_{\pm.14}$ | $\mathbf{0.47}_{\pm.28}$ |
| Small | 100k | $0.50_{\pm.00}$ | $\mathbf{0.72}_{\pm.21}$ | $0.72_{\pm.21}$ | $0.15_{\pm.01}$ | $\mathbf{0.40}_{\pm.37}$ | $0.32_{\pm.14}$ | $0.29_{\pm.04}$ | $0.40_{\pm.23}$ | $\mathbf{0.42}_{\pm.16}$ |
| Medium | 10k | $0.83_{\pm.23}$ | $1.01_{\pm.01}$ | $\mathbf{1.02}_{\pm.00}$ | $0.25_{\pm.12}$ | $\mathbf{0.84}_{\pm.02}$ | $0.84_{\pm.10}$ | $0.36_{\pm.22}$ | $0.71_{\pm.20}$ | $\mathbf{0.78}_{\pm.24}$ |
| Medium | 50k | $0.66_{\pm.21}$ | $1.01_{\pm.01}$ | $\mathbf{1.02}_{\pm.00}$ | $0.14_{\pm.04}$ | $0.85_{\pm.05}$ | $\mathbf{0.93}_{\pm.07}$ | $0.15_{\pm.04}$ | $0.45_{\pm.03}$ | $\mathbf{0.47}_{\pm.04}$ |
| Medium | 100k | $0.51_{\pm.00}$ | $\mathbf{1.02}_{\pm.00}$ | $\mathbf{1.02}_{\pm.00}$ | $0.14_{\pm.03}$ | $0.60_{\pm.24}$ | $\mathbf{0.97}_{\pm.02}$ | $0.22_{\pm.04}$ | $0.44_{\pm.01}$ | $\mathbf{0.53}_{\pm.10}$ |
| Large | 10k | $\mathbf{1.02}_{\pm.00}$ | $1.01_{\pm.00}$ | $1.01_{\pm.00}$ | $0.79_{\pm.22}$ | $0.87_{\pm.06}$ | $\mathbf{1.04}_{\pm.02}$ | $0.78_{\pm.14}$ | $0.84_{\pm.07}$ | $\mathbf{0.87}_{\pm.06}$ |
| Large | 50k | $\mathbf{1.02}_{\pm.00}$ | $1.01_{\pm.00}$ | $1.01_{\pm.00}$ | $0.68_{\pm.06}$ | $0.97_{\pm.05}$ | $\mathbf{1.00}_{\pm.01}$ | $0.47_{\pm.03}$ | $0.55_{\pm.11}$ | $\mathbf{0.57}_{\pm.13}$ |
| Large | 100k | $1.01_{\pm.00}$ | $1.01_{\pm.00}$ | $1.01_{\pm.00}$ | $0.73_{\pm.15}$ | $0.92_{\pm.06}$ | $\mathbf{0.99}_{\pm.01}$ | $0.39_{\pm.07}$ | $0.54_{\pm.04}$ | $\mathbf{0.74}_{\pm.27}$ |

**Quality of Latent Behavior Compression.** We also compared the policies encoded in the latent space with the ones in the training dataset. They were compared by examining the *performance recovery*, that is, the ratio between the performances of policies decoded from the latent space and those in the dataset. The values for MC are reported in Table 1.[4] First, it is clear that increasing the number of latent dimensions or policy size frequently leads to better performance recovery, resulting in higher performance as well. Interestingly, some configurations appear to recover *higher* performances than the ones in the training dataset. This may be due to the generalization abilities of the AE, but it may also be influenced by variance in the policy evaluation process. On the contrary, we note that 1D latent spaces trained on Small policies fail to learn any meaningful encoding of the behaviors, collapsing to a uniform representation. We attribute this phenomenon to the instability of the learning process when the latent dimensions are not sufficient. Interestingly, this issue is almost always fixed by increasing the number of latent dimensions and does not arise with large policies, which show excellent performance recovery. Finally, our analysis does not indicate that the dataset's dimension has any meaningful influence on performance recovery.

**Generalization.** We further investigated the generalization capabilities of Latent Behavior Compression in more complex environments. Focusing on larger latent spaces where grid visualization is infeasible, we estimate performance recovery via random sampling of the latent space instead. Table 2 reports the results for HC and HP. These experiments corroborate the observations in MC and provide statistically significant evidence that the latent space generalizes beyond the training set. More specifically, increasing the latent dimension consistently improves performance recovery in HC, while such trend is likely obscured in HP by the high variance in the sampling process. Finally, we observe that generalization varies by task complexity, as indicated by the lower recovery rates for difficult tasks like `frontflip` and `backflip` in HC.

**Takeaways.** With these experiments, we provided a positive answer to (**Q1**): the proposed unsupervised pipeline is indeed capable of encoding behaviorally meaningful policies in a wide range of configuration and in multiple environments, ultimately leading to a compression of up to five orders of magnitude.[5] As for (**Q2**), we found that while larger policies produce richer behavioral manifolds, even a one-dimensional latent space is often sufficient to capture a wide range of behaviors, supporting the hypothesis that the intrinsic dimensionality of the policy behavior manifold is dictated by the environment complexity rather than by the cardinality of the parameterization. Additionally, we study the scalability and generalization capabilities of the latent space. Finally, we extracted some evidence for the existence of a critical intrinsic dimension in the behavioral manifold, but how to leverage this evidence to learn the *best latent representation possible* is out of the scope of our work.

---

[4]Values related to the `left` task have been omitted as they do not present major differences from the `standard` task. Instead, they are reported in Table 3 of Appendix B.

[5]More precisely, a 121801:1 compression rate at peak.

Table 2: Quality of Latent Behavior Compression in HP and HC. We report the performance recovery for four tasks by the mean performance recovery and 95% confidence interval over 10 seeds. **Bold** indicates best per task.

| Environment | Task | Latent Dimensions | | | | | |
| --- | --- | --- | --- | --- | --- | --- | --- |
| | | 5D | | 8D | | 16D | |
| | | Mean | 95% CI | Mean | 95% CI | Mean | 95% CI |
| **Hopper** | `forward` | 1.33 | $[1.03, 1.69]$ | **1.59** | $[1.21, 1.99]$ | 1.48 | $[1.21, 1.78]$ |
| | `backward` | **2.66** | $[0.95, 5.96]$ | 1.29 | $[1.11, 1.47]$ | 1.20 | $[1.07, 1.33]$ |
| | `standstill` | 1.49 | $[1.06, 2.08]$ | 1.51 | $[1.02, 2.25]$ | **1.57** | $[1.07, 2.32]$ |
| | `jump` | **3.83** | $[0.64, 9.32]$ | 1.54 | $[1.07, 2.14]$ | 2.42 | $[1.35, 3.75]$ |
| **HalfCheetah** | `forward` | 1.63 | $[1.02, 2.31]$ | 1.80 | $[1.20, 2.47]$ | **1.84** | $[1.41, 2.39]$ |
| | `backward` | 1.27 | $[1.04, 1.55]$ | 1.52 | $[1.19, 1.85]$ | **1.72** | $[1.37, 2.06]$ |
| | `frontflip` | 0.54 | $[0.29, 0.86]$ | 0.74 | $[0.49, 0.99]$ | **1.20** | $[0.80, 1.63]$ |
| | `backflip` | 0.55 | $[0.31, 0.86]$ | 0.75 | $[0.51, 0.99]$ | **1.23** | $[0.81, 1.66]$ |

## 5.2 SUPERVISED FINE-TUNING

Finally, we address the last research question, namely:

> (**Q3**) How can we fine-tune against specific tasks, leveraging the low-dimensional space? Does this offer any advantages?

To achieve this, we compared the effect of supervised fine-tuning on the latent space using a simple variant of PGPE, which we refer to as Latent PGPE, with various baselines, including PPO (Schulman et al., 2017), DDPG (Lillicrap et al., 2016), TD3 (Fujimoto et al., 2018), and SAC (Haarnoja et al., 2018). We also tested PGPE (Sehnke et al., 2008) in the high-dimensional parameter space, referred to as Parameter PGPE, as a sanity check. Our implementation of both Parameter PGPE and Latent PGPE can be found in Appendix B. All the algorithms were run in the best-performing configuration for the policy sizes. Interestingly, DRL baselines always struggle to optimize small policies and typically perform better with larger ones. On the contrary, Parameter PGPE benefits from having a reduced set of parameters to control; however, it generally suffers from high sample complexity. This evidence is reported in Appendix B (Fig. 7,8,9).

From these comparisons (reported in Fig. 4 and Fig. 5 for MC and RC, respectively), we were able to extract three main findings. First of all, the convergence rate and performance of Latent PGPE are positively correlated with the number of dimensions of the latent space, which confirms that larger latent spaces are indeed better shaped and with an easier optimization landscape, as hinted in Subsec. 5.1 as well. Secondly, Latent PGPE converges faster than all baselines in 7 out of 8 tasks, even though it does not always converge to the optimum. Finally, Latent PGPE is able to achieve comparable, if not better, performances to most of the baselines, even for complex tasks like `speed` in Fig. 4c, 5a. However, we observe that it fails to solve the `height` task in Fig. 4d, due to the scarce representation of the high-performance policies in the unsupervised policy dataset.

**Takeaways.** These results provide a positive answer to (**Q3**): Leveraging the learned low-dimensional representation of the behavioral manifold, the agent does not only achieve faster convergence, but better performance than state-of-the-art DRL algorithms in challenging sparse tasks. Unfortunately, the trade-off is that the fine-tuning performance is tied to the quality of the learned representation.

## 6 CONCLUSIONS

We proposed a novel, unsupervised framework to address the sample inefficiency of Deep Reinforcement Learning by shifting the focus from parameter space to behavior space. Our approach successfully learns a compact latent manifold of policies, organized by behavioral similarity, using a generative model with an unsupervised behavioral reconstruction loss. Empirically, we showed that this approach can compress policy parameterizations by several orders of magnitude while preserving their functional expressivity. This compressed representation also allows for more efficient fine-tuning for downstream tasks via gradient-based optimization in the low-dimensional latent space.

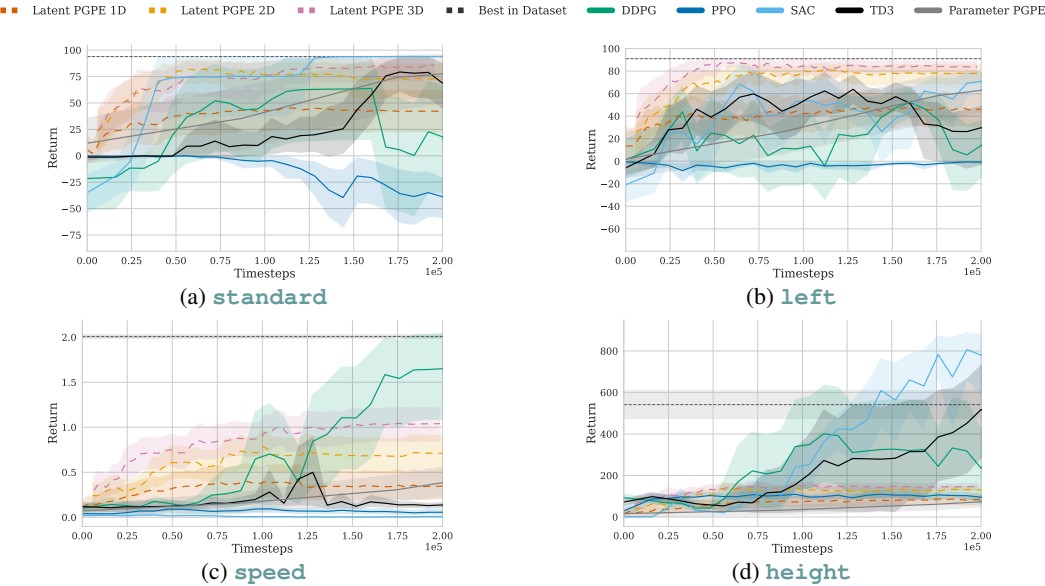

Figure 4: Performance comparison in MC for different tasks. We report the average and 95% confidence interval over 10 runs.

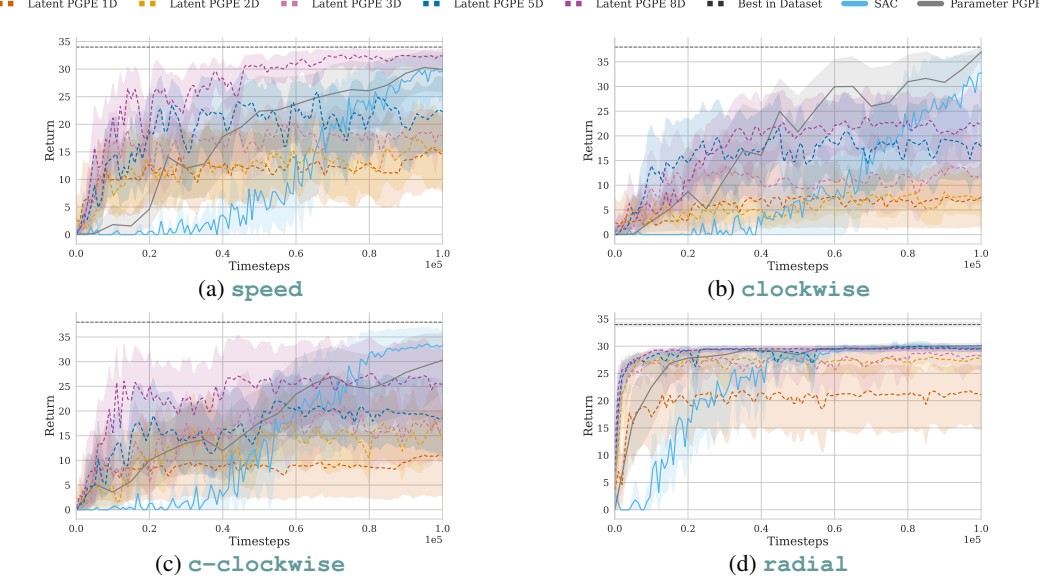

Figure 5: Performance comparison in RC for different tasks. We report the average and 95% confidence interval over 10 runs. For clarity, the worst-performing baselines are omitted. A full study is reported in Appendix B.

**Future Directions.** Our framework is intentionally modular, and we view it as a blueprint for a new class of more efficient DRL agents. We believe this approach can inspire significant future research into its core components, including the use of alternative behavioral divergences, more advanced generative architectures for compression, and the adaptation of different algorithms for latent behavior optimization. Furthermore, many RL approaches that condition value functions (Faccio et al., 2021) or meta-learners (Rakelly et al., 2019) directly on policy parameters could greatly benefit from our compression, as it provides a compact and semantically meaningful representation to replace raw parameter vectors.

## ETHICS STATEMENT

This work presents fundamental research in reinforcement learning theory and algorithms. We have carefully reviewed the Code of Ethics and confirm that our research raises no ethical concerns.

## REPRODUCIBILITY STATEMENT

All experiments were run on 50 cores of an Intel(R) Xeon(R) Gold 64118H CPU, 1 TB of RAM, and one NVIDIA H100 GPU. The total wall-clock time to re-run all experiments is approximately 50 hours. Our proposed approach is detailed in Section 4 with full implementation details and hyperparameters included in Appendix B. The source code is available in the supplementary material and at this link.

## LLM DISCLOSURE

We used LLMs in a limited capacity as a writing assistance tool. Specifically, LLMs were employed to help refine the clarity and readability of selected paragraphs throughout the paper. All content has been reviewed and verified by the authors, who take full responsibility for the accuracy and originality of all statements in this paper.

## ACKNOWLEDGMENTS

We thank Andrea Fraschini for his assistance in conducting the baseline experiments and Keenan Crane for the extremely clear visualization of autoencoder spaces used in Fig. 1 and that can be found at this link. This work was supported by the Italian Ministry of University and Research (MUR) under the National Recovery and Resilience Plan (NRRP), and by the European Union (EU) under the NextGenerationEU project (CUP D43C24001670008).

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

## A  RELATED WORKS

**Simplification of the Policy Space.** A variety of works (among others, Gregor et al., 2017; Eysenbach et al., 2018; Achiam et al., 2018; Hansen et al., 2019) have proposed methods to simplify the policy space. Yet, those policies should be generally intended as mere initializations for supervised fine-tuning, which falls back to operating in the original policy space once the downstream task is revealed. To the best of our knowledge, the only other work defining a formal criterion to operate a compression of the policy space is Mutti et al. (2022). Yet, this paper seeks a way to reduce the cardinality of the policy space, rather than its dimensionality. Moreover, the constraints defining a valid compression are stricter than ours, resulting in an optimization problem that is known to be NP-hard. Finally, their work does not provide a way to perform *supervised fine-tuning* in a scalable way.

**Weight Space Learning** The goal of our work, namely, to learn a latent representation of neural network parameters, is shared by the field of Weight Space Learning (WSL). In RL, recent works have investigated task-specific generative models, conditioning them on goals (Faccio et al., 2023) or performance checkpoints (Peebles et al., 2022), unlike our fully unsupervised approach. Our approach is more in line with the works on *hyper-representations* (Schürholt et al., 2021; 2022; 2024) of networks trained for supervised tasks. Hyper-representations are task-agnostic low-dimensional embeddings of neural networks learned from a zoo of trained models. Beyond its novel application to RL, our approach differs from these works in two ways. First, our "zoo" does not require trained experts. Second, and more critically, WSL methods must often contend with the vast number of parameter-space symmetries (e.g., neuron permutations and scaling (Kůrková & Kainen, 1994)) using complex architectures or workarounds. Our behavioral reconstruction loss fundamentally sidesteps this challenge. By compressing policies based on their function rather than their specific parameters, our autoencoder naturally assimilates these redundant parameterizations. This concept of functional compression is similar to "policy fingerprinting" (Faccio et al., 2023), which also gauges behavior independent of weights.

**Policy Manifold and Quality Diversity.** The idea of employing generative models to learn a compressed representation of the policy space has received some recent attention outside of the Weight Space Learning field. Rakicevic et al. (2021) hypothesized that there might be a low-dimensional manifold embedded in the policy parameter space, even if they did not characterize it formally. Chang et al. (2019) trains a Variational AE to reconstruct the weights of pre-trained expert policies to learn expert-agent embeddings and analyze the latent structure of the solution space. A similar architecture has also been applied in the field of Quality Diversity, either to improve the sample efficiency of diversity-based search Rakicevic et al. (2021) or to distill a large policy archive into a compact generative model (Hegde et al., 2023). Notably, all the methods above employ VAE architectures with a *parameter-reconstruction loss*, which allows only moderate compression ratios of up to $19:1$ (Hegde et al., 2023). In comparison, this paper introduces a fully unsupervised pipeline that focuses on compressing a behavioral loss, intending to provide a compact space for latent policy optimization *as well* while retaining greater compression abilities.

**Policy Optimization.** First-order methods have been extensively employed to address PO (Peters & Schaal, 2008; Lillicrap et al., 2016) as well as natural gradients (Kakade, 2001) and trust-region methods (Schulman et al., 2017). Yet in this work, we built on the long tradition of PGPE Algorithms (Sehnke et al., 2008; Rückstiess et al., 2010; Miyamae et al., 2010; Montenegro et al., 2024), as their hard scalability to large parameter spaces is notoriously a blocking factor. Finally, we notice that Rakicevic et al. (2021) indeed proposed a method to optimize the diversity of the policies by taking into account the Jacobian of the decoder in a VAE architecture.

## B  EXPERIMENTAL DETAILS

**Environments.** We evaluate our methods on four control environments from the Gymnasium library: Mountain Car Continuous, Reacher, Hopper, and HalfCheetah. The first is a classic control environment, which consists of a car placed stochastically in the middle of a sinusoidal valley, with the goal state on top of the right hill. The state is defined by two continuous variables: the position of the car along the $x$-axis $p \in [-1.2, 0.6]$, and the velocity of the car $v \in [-0.07, 0.07]$. The only possible action is to apply an acceleration $a \in [-1, 1]$ to the car. The `standard` task is defined as $R_{\text{standard},t} = -0.1a^2$, until the goal is reached and a reward of $R_{\text{standard},t} = 100$ is obtained, and

the episode ends. If the goal is not reached, the episode ends after 999 steps. We introduce three additional tasks: **left**, which is the same as the standard task, but with the goal moved to the top of the left hill ($p \leq -1.1$); **height**, which gives a reward of $R_{\text{height},t} = h^2$ at each time step for which $h \geq 0.2$, with $h = \sin(3p) * 0.45 + 0.55$ being the height of the car; **speed**, which gives a reward of $R_{\text{speed},t} = v^2$ at each time step. In the **left** task, the episode ends when the car reaches the left goal, while in the **height** and **speed** tasks, the episode ends when the car reaches the right goal.

The second environment, Reacher, is a classic continuous control task consisting of a two-jointed robot arm, moving in a 2D space, with an end-effector called *fingertip*. The state is originally 10-dimensional, but we remove the coordinates of the target and the vector between the fingertip and the target. We end up with a 6-dimensional state composed of: $\cos(q_1), \cos(q_2), \sin(q_1)$, and $\sin(q_2)$, the cosines and sines of the two joint angles, and $\omega_1$ and $\omega_2$, their angular velocities. For the purpose of normalization, we consider the state bounded between the vectors $[-1, -1, -1, -1, -5, -5]$ and $[1, 1, 1, 1, 5, 5]$. The agent controls the arm by applying a distinct torque to each hinge, making the action space $\mathcal{A} = [-1, 1]^2$. We disregard the standard task and instead define four new behavioral tasks that have the same reward shape $R_{\text{task}} = 1$ if the condition is met, or 0 otherwise. In the **speed** task, the condition is that the linear velocity of the tip is greater than 6. In the **clockwise** and **c-clockwise** tasks, the condition is that the tangential velocity of the fingertip is greater than -11, or 1, respectively. Finally, in the **radial** task, the condition is that the radial velocity of the tip is greater than 3. The episodes terminate after 50 steps.

The third environment, Hopper, is a classic continuous control task that models a 1-legged robot with 3 joints. The state space is 11-dimensional, capturing the robot's $z$-position, joint angles, and corresponding velocities. For the purpose of normalization, we consider the state bounded between the vectors $[0.7, -0.2, -2.7, -2.7, -0.8, -5.0, -5.0, -5.0, -5.0, -5.0, -5.0]$ and $[1.5, 0.2, 0.0, 0.0, 0.8, 5.0, 5.0, 5.0, 5.0, 5.0, 5.0]$. The agent controls the arm by applying a distinct torque to each hinge, making the action space $\mathcal{A} = [-1, 1]^3$. Analogously to Reacher, we disregard the standard task and instead define four new behavioral tasks that have the same reward shape $R_{\text{task}} = 1$ if the condition is met, or 0 otherwise. In the **forward**, **backward**, and **standstill** tasks, the condition is that the $x$-axis velocity is greater than 1, lower than -1, or between -0.05 and 0.05. In the **jump** task, the $z$ position has to be greater than 1.3. The episodes terminate after 1000 steps.

The fourth environment, HalfCheetah, is a classic continuous control task that models a 2D, two-legged robot with 6 actuated joints. The state space is 18-dimensional, including the robot's body position, rotational angles, and joint velocities. For the purpose of normalization, we consider the state bounded between the vectors $[0, -\pi, -0.52, -0.785, -0.4, -1, -1.2, -0.5, -5.0, -5.0, -5.0, -5.0, -5.0, -5.0, -5.0, -5.0, -5.0]$ and $[1.5, \pi, 1.05, 0.785, 0.785, 0.7, 0.87, 0.5, 5.0, 5.0, 5.0, 5.0, 5.0, 5.0, 5.0, 5.0, 5.0]$. The agent controls the arm by applying a distinct torque to each hinge, making the action space $\mathcal{A} = [-1, 1]^6$. The tasks are almost the same as for Hopper. In the **forward** and **backward**, the condition is that the $x$-axis velocity is greater than 2, or lower than -2. In the **frontflip** and **backflip**, the task condition is that the cumulative rotation along the $y$-axis increases or decreases by a full rotation from the initial angle. The episodes terminate after 1000 steps.

**Policies.** To model the policies, we use fully-connected, feed-forward, deterministic MLPs. Our choice of focusing on deterministic policies is dictated by the use of PGPE as an optimization algorithm in the last stage; however, our pipeline is designed to be general. As such, we believe there is no apparent limitation to the use of stochastic policies instead. The input layer has $\mathcal{S}$ neurons and is preceded by a normalization layer that standardizes the state features to have zero mean and unit variance. The hidden linear layers are followed by `elu` nonlinearities. The last layer has $|\mathcal{A}|$ neurons, followed by a `tanh` activation to squash the action into the valid range. We test three different shapes of policies in Mountain Car: Small policies composed of a single 4-neuron hidden layer; Medium policies composed of two 32-neuron hidden layers; Large policies composed of a 400-neuron hidden layer followed by a 300-neuron hidden layer. The number of parameters of the policies increases roughly by two orders of magnitude at each interval ($P_{\text{Small}} = 17$, $P_{\text{Medium}} = 1,185$, $P_{\text{Large}} = 121,801$). For Hopper and HalfCheetah, we use the same shape of Medium policies. While in Reacher we use Medium policies composed of two 64-neuron hidden layers, with $P_{\text{Reacher}} = 4738$.

**Policy Divergence.** To compute the divergence between policies, we instead estimate the distance of the deterministic actions over a subset of states. We consider the state spaces bounded as previously

described, and we extract roughly $M = 3000$ states. In Mountain Car, we find them by discretizing the two dimensions and creating a grid, while in MuJoCo environments, we simply sample them uniformly from the bounded state space. In the $k_n$-NN phase, we use $k_n = 15$, and compute the distance between two policies as:

$$D(\pi_{\boldsymbol{\theta}} \parallel \pi_{\boldsymbol{\theta}'}) = \sqrt{\sum_{i=1}^{M} \|\pi_{\boldsymbol{\theta}}(s_i) - \pi_{\boldsymbol{\theta}'}(s_i)\|_2^2}.$$

While in the manifold learning phase, we compute it as:

$$D(\pi_{\boldsymbol{\theta}} \parallel \pi_{\boldsymbol{\theta}'}) = \frac{1}{M}\sum_{i=1}^{M} \|\pi_{\boldsymbol{\theta}}(s_i) - \pi_{\boldsymbol{\theta}'}(s_i)\|_2^2.$$

**Autoencoder.** We use a simple, fully-connected, feed-forward, deterministic MLP to model the autoencoder. The shape of the autoencoder is the same for all the experiments. The input and output layers have size $P$, with the input layer being preceded by a standardization layer, and the output layer not being activated; the encoder has a 25-neuron hidden layer followed by a 10-neuron hidden layer; the decoder has the mirrored shape of the encoder. The first layer of the encoder and the first layer of the decoder are followed by `elu` nonlinearities. The autoencoder is trained for 50 epochs using the Adam optimizer with an initial learning rate of 0.0001 and a batch size of 64. We employ a learning rate scheduler that halves the learning rate after 15 epochs of non-improvement, evaluated on a 20% random hold-out set. The empirical loss used to train the autoencoder is defined as:

$$\mathcal{L}_B = \frac{1}{N}\frac{1}{M'}\sum_{i=1}^{N}\sum_{j=1}^{M'} \left\|\pi_{\boldsymbol{\theta}_i}(s_j) - \pi_{\hat{\boldsymbol{\theta}}_i}(s_j)\right\|_2^2,$$

where $N$ is the number of policies in the training dataset, $M' = 1000$ is the size of the subset of the state set that we sample at each gradient step, and $\pi_{\hat{\boldsymbol{\theta}}_i}$ is the reconstructed policy. In Mountain Car, we set the latent dimension of the autoencoders to $k = 1, 2, 3$, while in Reacher, we use $k = 1, 2, 3, 5, 8$. In Hopper and HalfCheetah, $k = 5, 8, 16$.

When we evaluate a latent space, we first compute the interquartile range for each dimension based on the spread of the training codes. Then, we discretize each dimension by a variable number of points depending on the dimension of the latent code: 100 points for 1D, 50 points for 2D, and 17 points for 3D. For more than three latent dimensions, we sample 10000 policies from the same bounded latent space at random. The decision is based solely on computational feasibility and serves the purpose of having a rough conservative estimate of the range of encoded behaviors.

**Performance Recovery.** When comparing the policies found in the latent space with the ones belonging to the original dataset, we compute a behavior recovery metric in the following way. First, we average the dataset lower and upper bounds for all tasks across three seeds with the same configuration, $lb_D, ub_D$. Then we do the same for the discretized set of policies reconstructed from the latent space, $lb_L, ub_L$. Finally, for each task, we compute the *performance recovery* as $\frac{ub_L - lb_D}{ub_D - lb_D}$. In Table 3, we provide the analysis for the reward **left**, which was omitted in Table 1.

**(Latent) PGPE.** As a byproduct of the low-dimensionality of the latent space, this framework is well suited to parameter-exploring PG methods. Algorithms like PGPE struggle with a high-dimensional set of parameters, such as those of a standard DRL network with hundreds of thousands of parameters. Yet, they can instead operate on the low-dimensional set of latent parameters while maintaining the expressivity of the original parameter space. As a bonus, the extension of PGPE to the latent space does *not* require computing the Jacobian of the decoder as in Eq. 5, as it can be seen as a deterministic addition to the black-box process that evaluates the parameters produced by the Gaussian hyperpolicy $\nu_{\boldsymbol{\phi}}$, where $\boldsymbol{\phi} = (\boldsymbol{\mu}, \boldsymbol{\sigma})$ is the vector of means and standard deviations parameterizing the Gaussian distributions over the latent parameters. In fact, the objective defined in Eq. 2 can be rewritten under the latent PG formulation as $J^R(\boldsymbol{z}) = \mathbb{E}_{\tau \sim p(\cdot|\boldsymbol{z}), \boldsymbol{z} \sim \nu_{\boldsymbol{\phi}}}[R(\tau)]$, with the only change being that the probability density of the trajectories is given by the policy induced by the latent parameters as $p(\tau \mid \boldsymbol{z}) = \mu(s_0)\prod_{t=0}^{T} \mathbb{P}(s_{t+1} \mid s_t, a_t)\, \pi_{g_{\boldsymbol{\varsigma}(\boldsymbol{z})}}(a_t \mid s_t)$. Finally, the gradient estimator at Eq. 3 is left unchanged, but for the change in parameter space from $\boldsymbol{\theta}_i$ to $\boldsymbol{z}_i$.

Table 3: Quality of Latent Behavior Compression in MC. We report the performance recovery for the **left** task. We report mean and standard deviation computed over 3 seeds.

| Config. | | Left | | |
|---|---|---|---|---|
| Policy | Dataset | 1D | 2D | 3D |
| Small | 10k | $0.73_{\pm.16}$ | $0.66_{\pm.18}$ | $0.98_{\pm.03}$ |
| | 50k | $0.64_{\pm.19}$ | $0.98_{\pm.03}$ | $0.99_{\pm.02}$ |
| | 100k | $0.73_{\pm.05}$ | $0.80_{\pm.21}$ | $0.94_{\pm.06}$ |
| Medium | 10k | $0.95_{\pm.05}$ | $1.01_{\pm.01}$ | $1.01_{\pm.01}$ |
| | 50k | $0.78_{\pm.13}$ | $1.01_{\pm.00}$ | $1.01_{\pm.00}$ |
| | 100k | $0.82_{\pm.10}$ | $1.01_{\pm.00}$ | $1.01_{\pm.01}$ |
| Large | 10k | $1.01_{\pm.00}$ | $1.00_{\pm.01}$ | $1.01_{\pm.00}$ |
| | 50k | $1.01_{\pm.00}$ | $1.01_{\pm.00}$ | $1.00_{\pm.00}$ |
| | 100k | $1.01_{\pm.00}$ | $1.01_{\pm.00}$ | $1.00_{\pm.00}$ |

We base our implementation of PGPE on an ask-and-tell implementation with symmetric sampling (Toklu et al., 2020). We modify it to allow for numpy parallelization, reward normalization, center learning rate scheduling, learning $\log \sigma$ instead of $\sigma$, and natural gradient computation. The center is optimized with Adam, with momentum $0.2$. The log-standard deviation instead is learned through simple gradient ascent with fixed learning rate. For Mountain Car, we perform 75 seeded runs on 75 different autoencoders with the same hyperparameters: center learning rate 0.05, population size 4, initial standard deviation 0.6, standard deviation learning rate 0.1, and 50 generations. In Reacher, the learning rate has a linear annealing down to 20% of the initial value, and we use the same hyperparameters for all runs, which are the following: center learning rate 0.1, population size 10, initial standard deviation 0.3, standard deviation learning rate 0.1, and 200 generations. In both cases, each sample of the population is evaluated on a single episode.

Given the episodic nature of PGPE, each generation can take an arbitrary number of environment steps to evaluate the samples of the population. Since the center is changed only once at the end of a generation, we evaluate the learning curve at different sample checkpoints, contrary to the standard StableBaselines3 approach. This causes different runs to have different evaluation checkpoints ($x$-axis of the learning curve) and a different number of total samples used. To visualize an aggregate learning curve across multiple runs, we take two measures: we interpolate the various curves along the $x$-axis; we extend the final evaluation of each run along the $x$-axis so that all runs have identical lengths, with the assumption that the algorithm has already converged by the end of the training. To ensure the stability of the final evaluation, we evaluate the center solution on 100 episodes (instead of 10 episodes used during training).

**Reproducibility.** In MC, we perform two main experiments. First, we study different configurations by creating 27 different datasets. We seed all the steps of the pipeline with seeds 0 through 26. In order, we use seeds 0-8 for Small policies, 9-17 for Medium policies, and 18-26 for Large policies. In each batch, the first three seeds are used for datasets of 10k policies, the next three for datasets of 50k, and the last three for datasets of 100k policies. The second experiment focuses on datasets of 10k Medium policies, and it is run with seeds starting from 100. In Reacher, we focus on datasets of 100k Medium policies with seeds starting from 0. In Hopper and HalfCheetah, we focus on datasets of 10k Medium policies with seeds starting from 0.

### ADDITIONAL EXPERIMENTS

**Baseline Hyperparameters.** Here we provide the hyperparameters used to train the baselines for each environment. Where not specified, we use the StableBaselines default parameters. In MC, we used higher standard deviations for the stochastic processes used by TD3 and DDPG for exploration: 0.75 and 0.65, respectively. For DDPG, we also used a smaller replay buffer size of 50000. For SAC, we used a soft update coefficient (tau) of 0.01, train frequency of 32, entropy coefficient of 0.1, 32 gradient steps per rollout, replay buffer size of 50000, and we used generalized State Dependent Exploration (gSDE). For PPO, we used a learning rate of 0.0001, 32 steps per rollout, batch size of 256, 4 epochs of optimization of the surrogate loss, lambda value of 0.9 for the Generalized Advantage Estimator (GAE), a clip range of 0.1, entropy coefficient of 0.1, value function coefficient

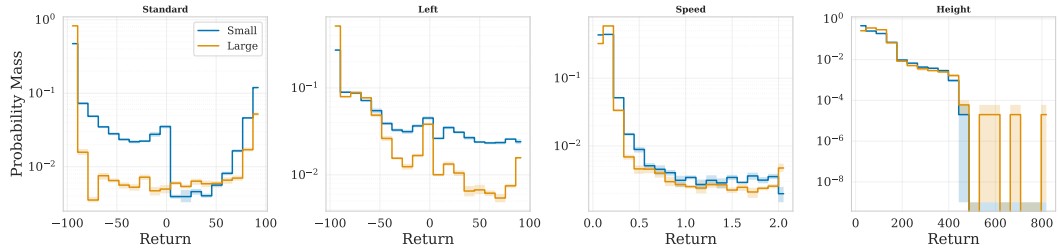

Figure 6: Reward distribution comparison of datasets of 10k Small (blue) and Large (orange) policies in MC. The $y$-axis uses a logarithmic scale. We report the average and 95% CI over 5 seeds.

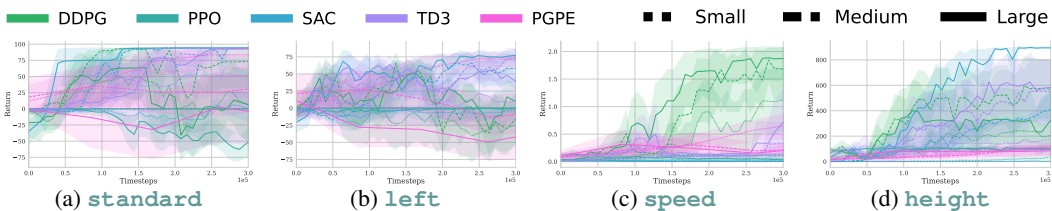

(a) `standard`  (b) `left`  (c) `speed`  (d) `height`

Figure 7: Baseline Ablation study in MC. We report the average and 95% confidence interval over 10 runs.

of 0.19, max gradient norm of 5 and we used gSDE. In RC, we kept the same hyperparameters used for MC, changing only the standard deviation of DDPG to 0.5.

**Baseline Ablation Study.** We report complete baseline studies for both MC and RC. In MC, we study how the baselines operate with different-sized policies. We report our results in Figure 7. We observe that almost all algorithms struggle with optimizing small policies. In Figure 8, we report a separate study for PGPE, in order to offer a cleaner visualization, given the major difference in sample complexity. We can observe the opposite behavior, namely, PGPE is often more sample-efficient and better-performing when using smaller policies. Finally, in Figure 9, we report the complete study of baselines for the RC environment.

**Policy Representation Range.** We analyze the behavioral distributions of randomly sampled Small and Large policies in the MC environment (Figure 8). For simpler tasks (`standard`, `left`, and `speed`), both architectures cover a similar reward range, though their probability distributions differ. However, in the more complex `height` task, Large policies exhibit a significantly broader support, reaching higher maximum rewards than their Small counterparts. While this observation relies on random weight sampling and is limited to a single environment, it aligns with theoretical findings on the relationship between parameter count and network expressivity (Montúfar et al., 2014; Raghu et al., 2017; Bahri et al., 2024), suggesting that larger networks are naturally capable of representing a richer diversity of behaviors.

**Latent Behavior Manifolds.** To complement the results in the main paper, we provide an extensive set of visualizations of the learned latent behavior manifolds. These plots illustrate how the latent representations organize policies across different tasks, policy sizes, and encoding dimensions. They cover both environments studied in this work—Mountain Car (MC) and Reacher (RC)—and show how the manifold structure emerges consistently across settings. The visualizations serve two purposes: (i) to confirm that the latent space captures meaningful behavioral structure qualitatively, and (ii) to demonstrate the consistency of this organization across seeds and settings. For clarity in the main text, we only reported a subset of representative plots; here, to enable a more thorough inspection and reproducibility, we visualize one seed per configuration in MC and all seeds for RC. Each figure shows the latent spaces for 1, 2, and 3 dimensions for all tasks. The tasks are ordered from left to right as follows: `standard`, `left`, `speed`, `height` for MC, and `speed`, `clockwise`, `c-clockwise`, and `radial` for RC.

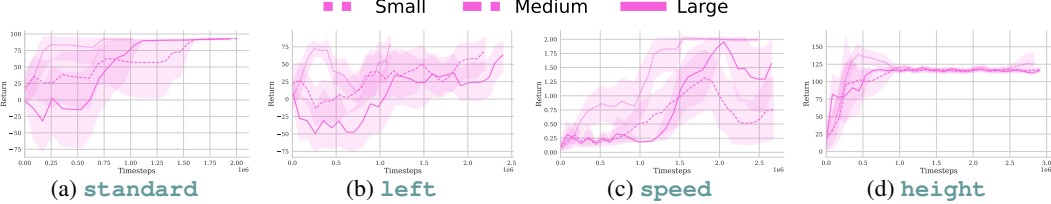

Figure 8: PGPE Ablation study in MC. We report the average and 95% confidence interval over 10 runs.

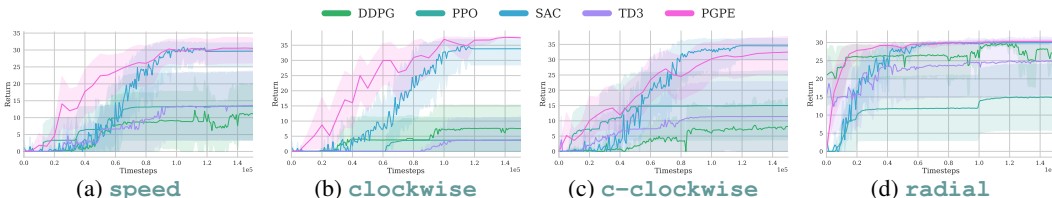

Figure 9: Baseline study in RC. We report the average and 95% confidence interval over 10 runs.

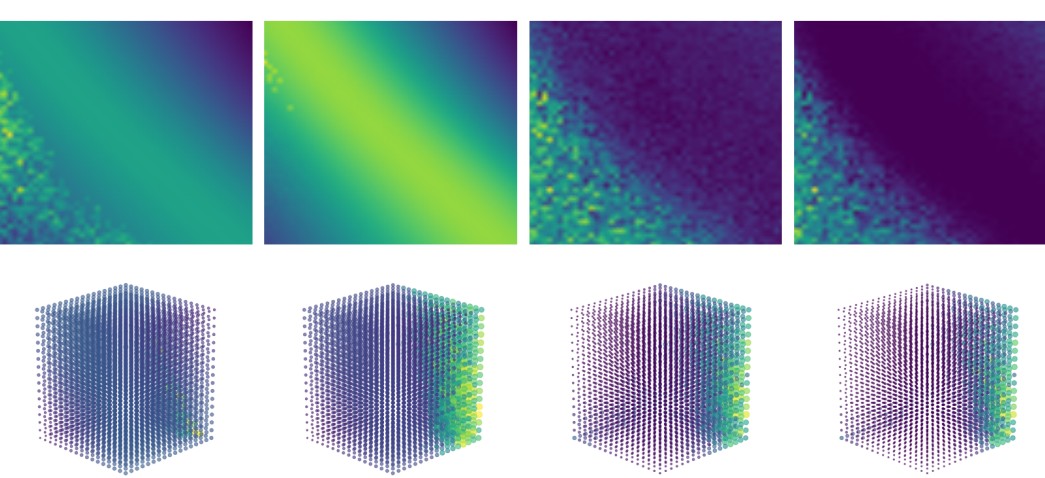

Figure 10: MC - Small, 10k - Seed 0

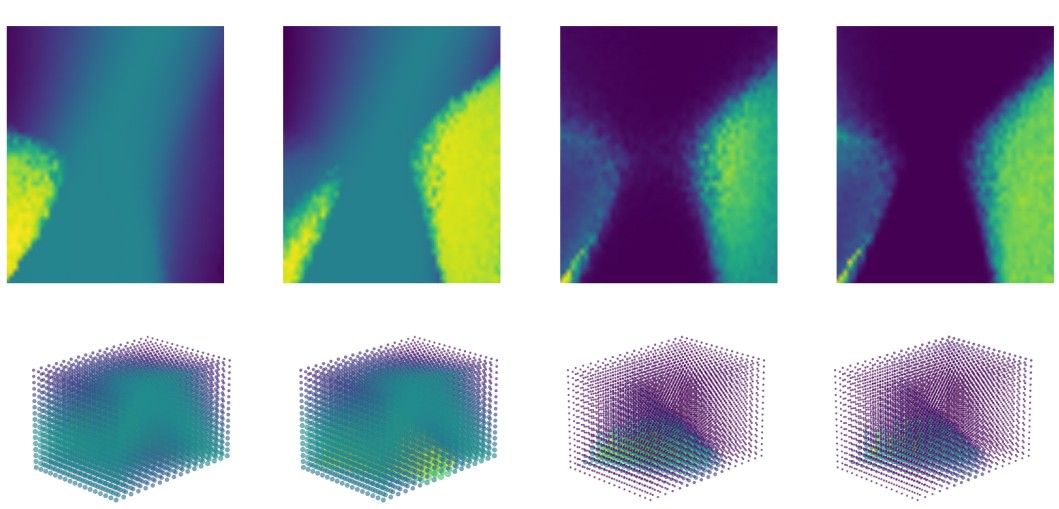

Figure 11: MC - Small, 50k - Seed 3

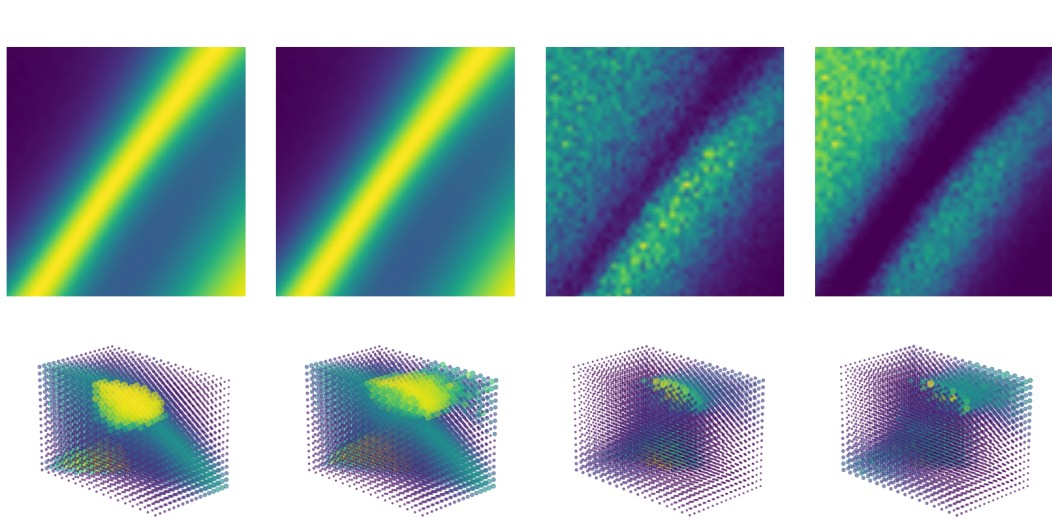

Figure 12: MC - Small, 100k - Seed 6

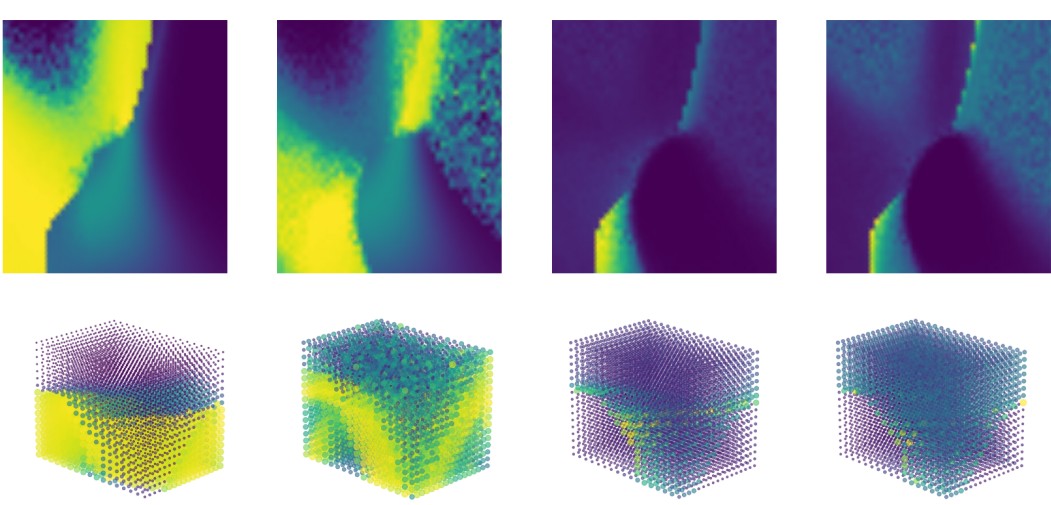

Figure 13: MC - Medium, 10k - Seed 9

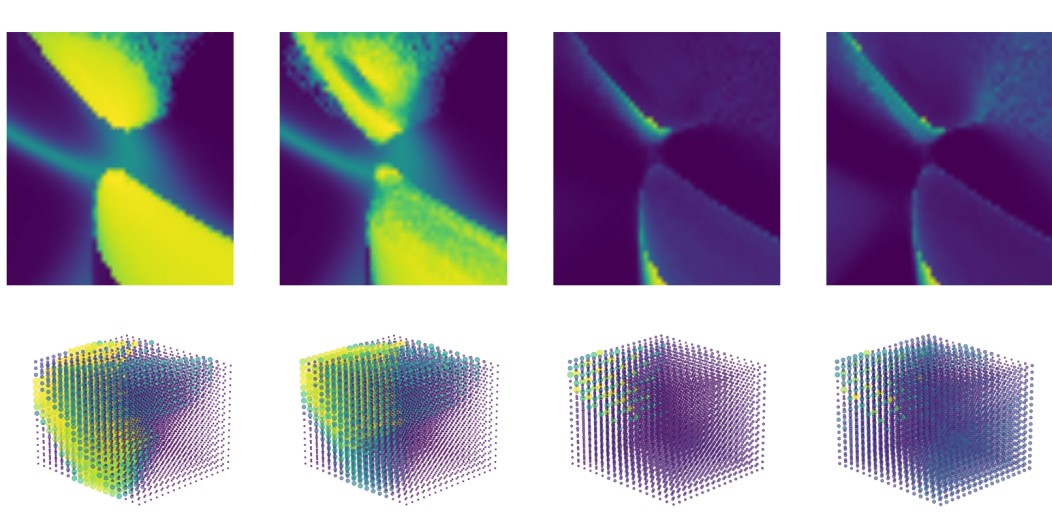

Figure 14: MC - Medium, 50k - Seed 12

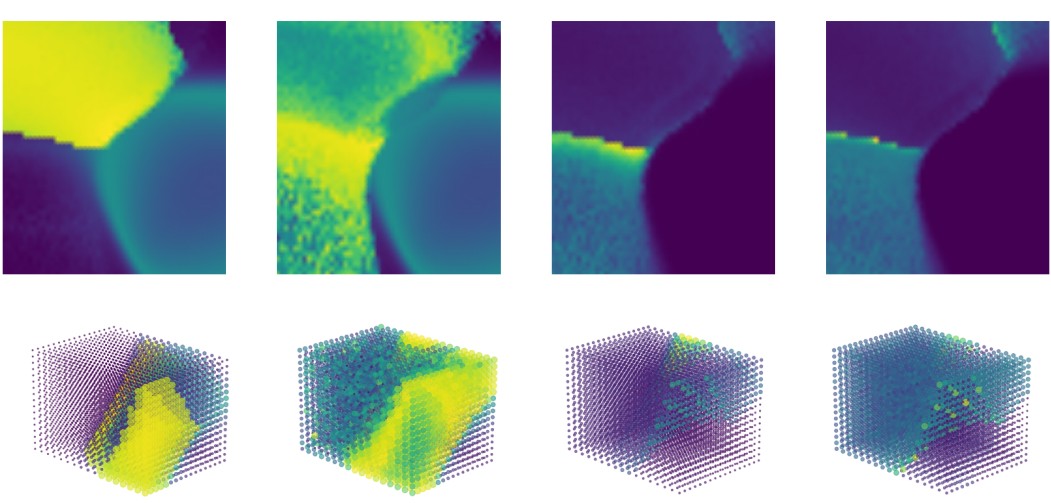

Figure 15: MC - Medium, 100k - Seed 15

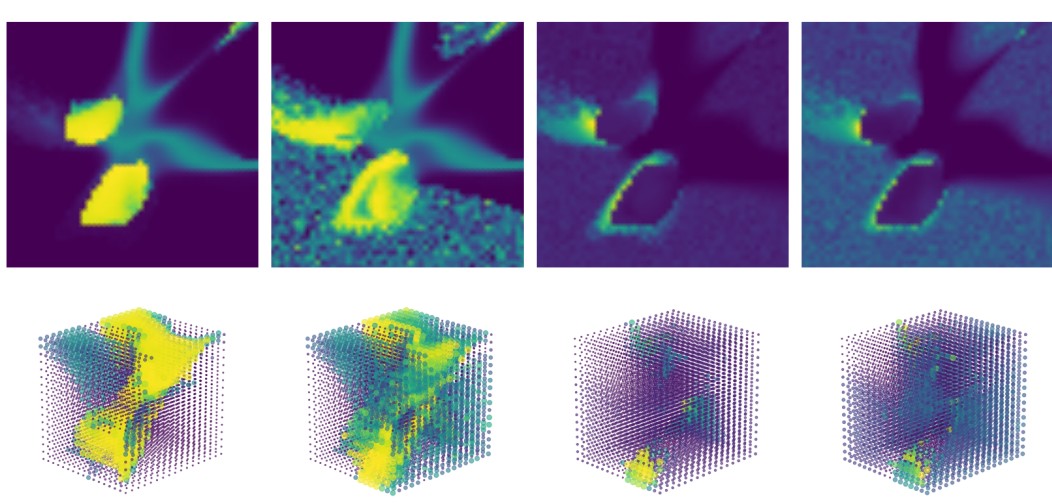

Figure 16: MC - Large, 10k - Seed 18

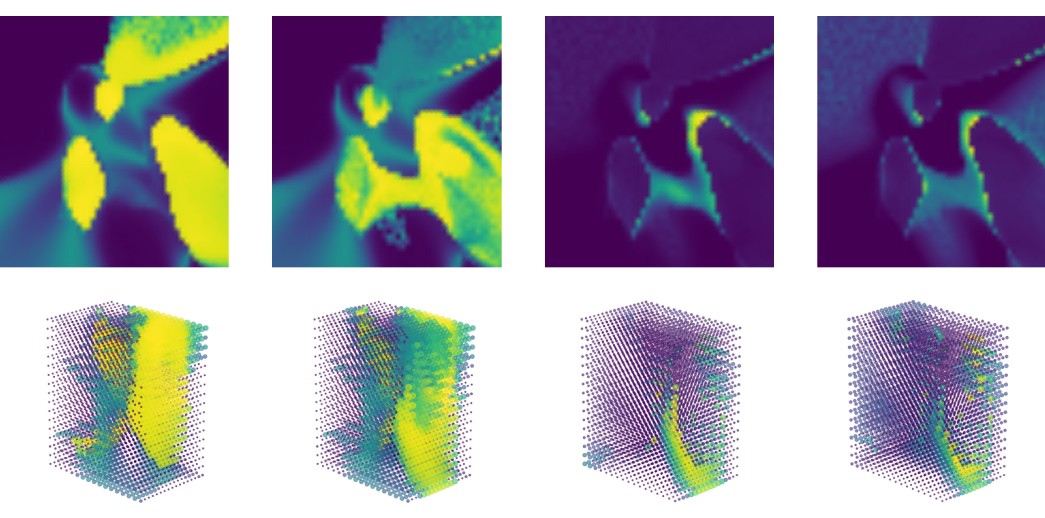

Figure 17: MC - Large, 50k - Seed 21

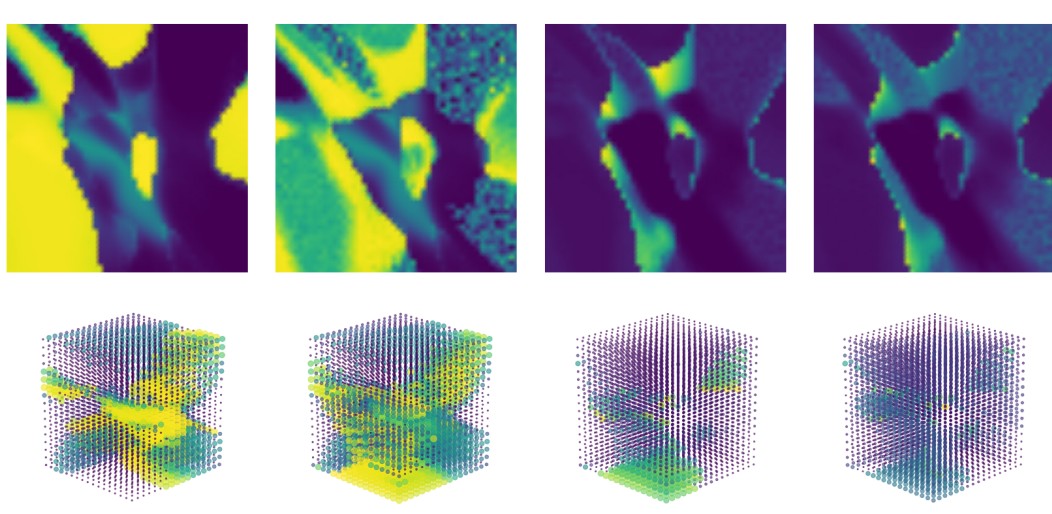

Figure 18: MC - Large, 100k - Seed 24

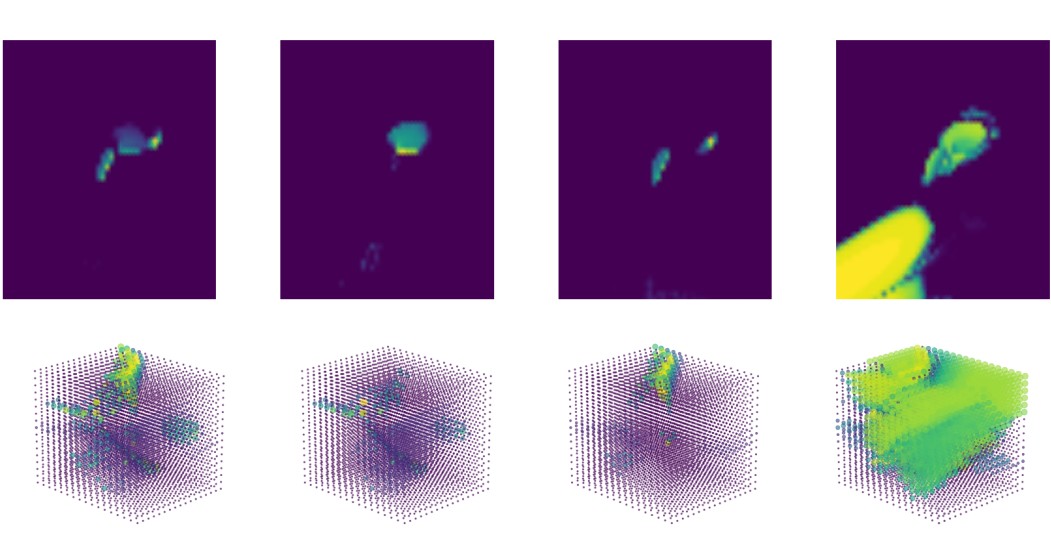

Figure 19: RC - Seed 0

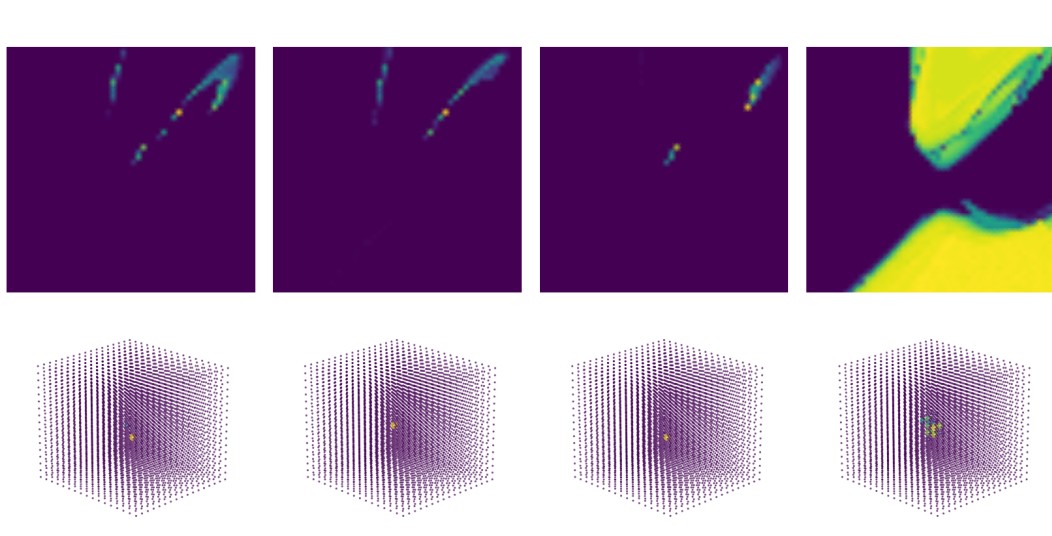

Figure 20: RC - Seed 1

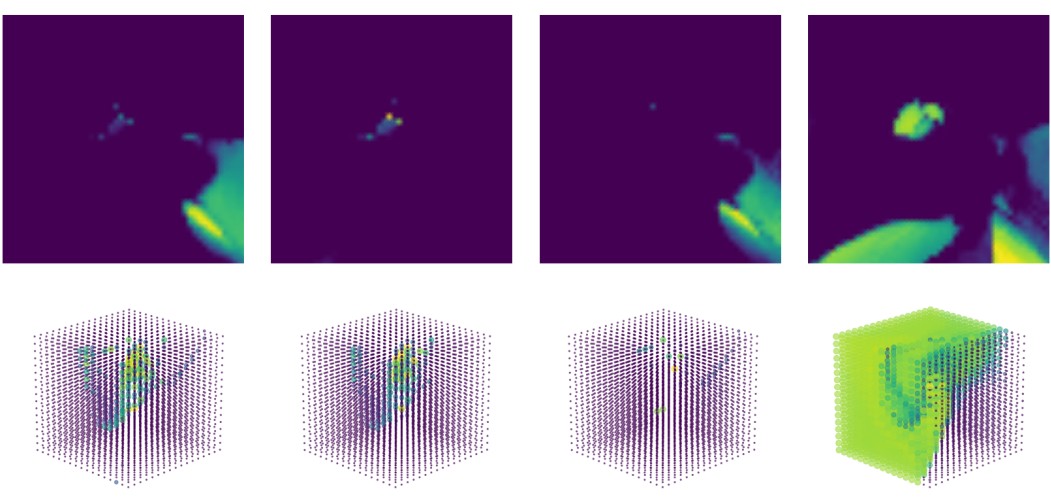

Figure 21: RC - Seed 2

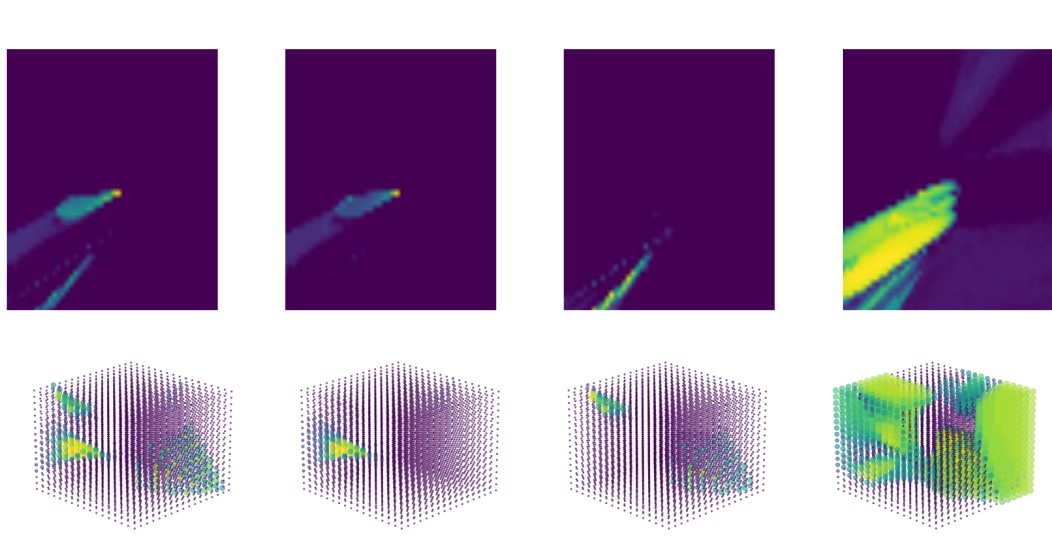

Figure 22: RC - Seed 3

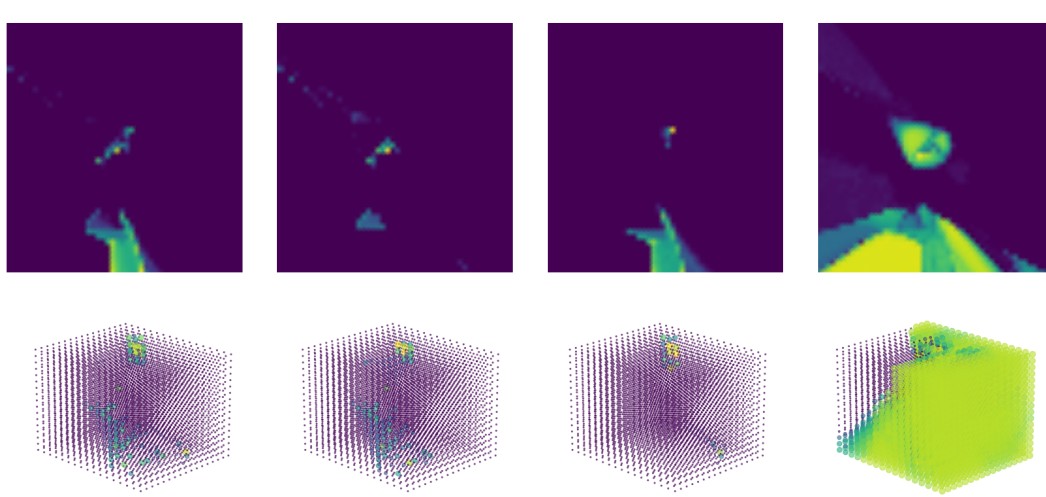

Figure 23: RC - Seed 4

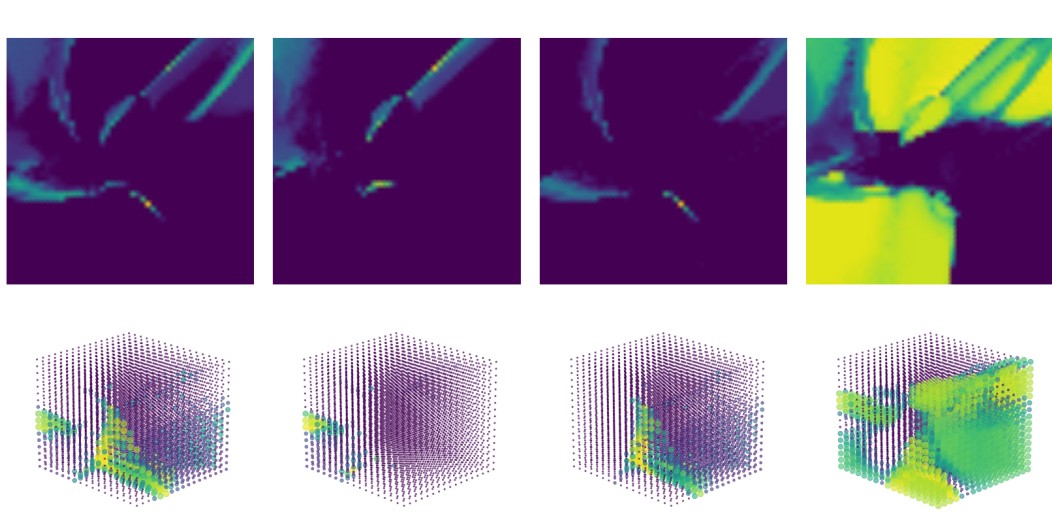

Figure 24: RC - Seed 5

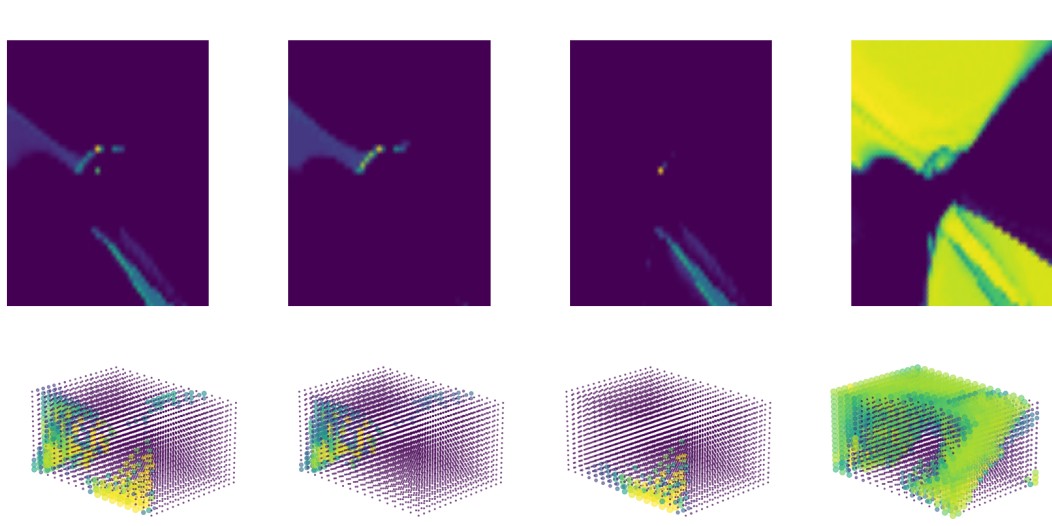

Figure 25: RC - Seed 6

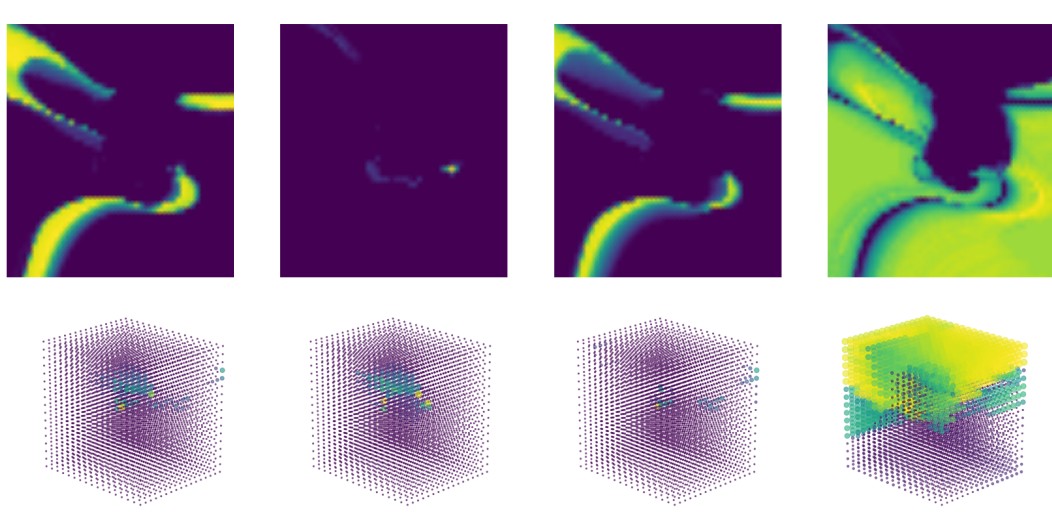

Figure 26: RC - Seed 7

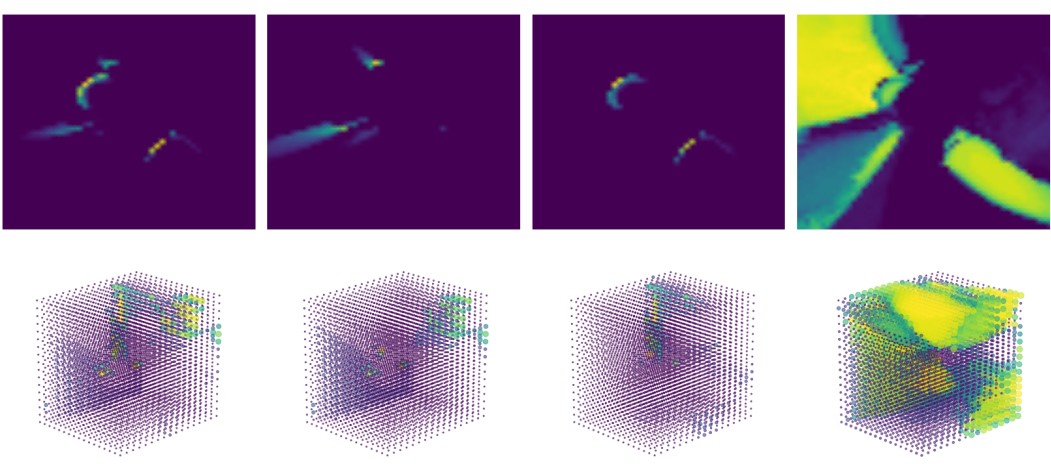

Figure 27: RC - Seed 8

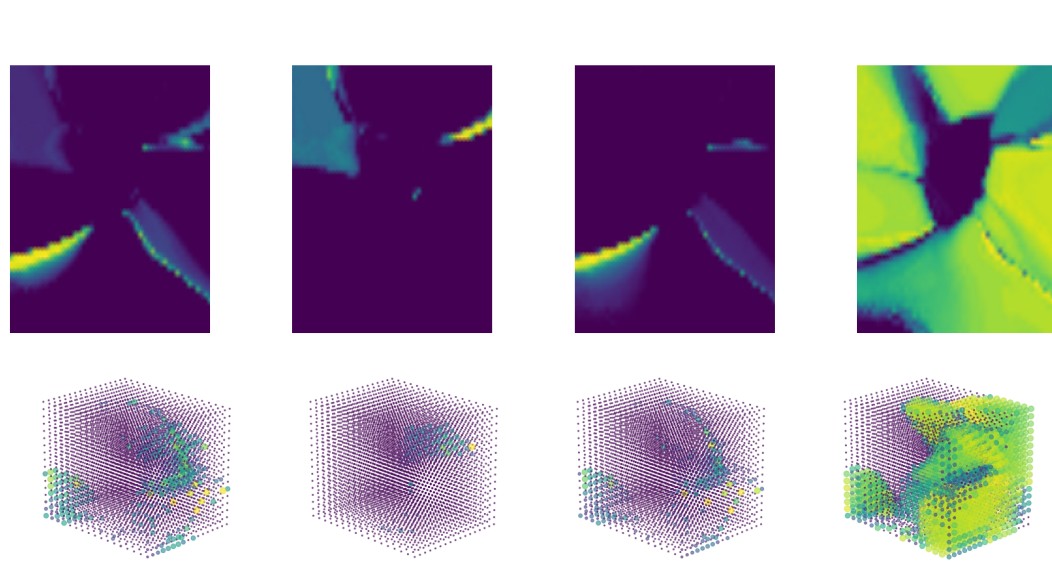

Figure 28: RC - Seed 9

