# OpenReview forum: "From Parameters to Behaviors: Unsupervised Compression of the Policy Space"
_ICLR.cc/2026/Conference — ICLR 2026 Poster_

### Official Review · Reviewer_uJZc · 2025-10-28

**Soundness:** 2
**Presentation:** 3
**Contribution:** 2
**Rating:** 6
**Confidence:** 4

**Summary:**

The paper tackles the issue of sample inefficiency in deep reinforcement learning caused by the high-dimensional parametrization of policies. The authors propose a two-stage approach: first, they compress the policy parameter space into a low-dimensional behavioral latent space using autoencoders; then, they leverage this learned latent representation to efficiently adapt and learn new tasks.

**Strengths:**

1. The paper provides a clear motivation, grounded in the Manifold Hypothesis, for learning and operating within a low-dimensional behavioral manifold.
2. The work addresses the sample inefficiency of DRL. At the pre-training stage, it uses unsupervised RL, which does not require one to solve actual RL tasks
3. The experiments show that the policy networks can be compressed up to five orders of magnitude
4. The experiments show that fine-tuning the latent space allows faster convergence and low sample complexity

**Weaknesses:**

My main concern is that although the work is well motivated and supported by good intuition, the empirical evidence provided in the experiments is not sufficiently strong to support some of the authors’ claims. In particular:

1. Complexity of the latent space: The authors evaluated latent space complexity through visualizations of the return of the encoded policies. However, the interpretation of these figures, especially for the 3D latent space, is somewhat ad hoc and lacks quantitative rigor. It would strengthen the paper to include concrete metrics that quantify latent space complexity rather than relying solely on visual inspection.

2. Performance recovery metric: To assess performance recovery, the authors compare the upper and lower performance bounds of policies found in the latent space versus those from the original datasets. In my view, this comparison is insufficient to convincingly demonstrate recovery, as it only reflects the spread of the distribution and neglects the distribution shape or central tendency. A more informative metric (such as divergence measures) could provide a clearer picture of how well the latent policies capture the original performance characteristics.

3. Convergence behavior: While Figures 4 and 5 show that Latent PGPE converges faster than other baselines, it does not consistently reach the optimal policy—approximately half of the time (e.g., Figures 4a, 4c, 4d, 5b, and 5c), convergence is suboptimal. In 5b, Parameter PGPE is significantly better than latent PGPE. This suggests that the chosen latent space may not be sufficiently expressive to fully recover the original policy space. This observation also reinforces my earlier point that spread-based metrics may fail to adequately capture true performance recovery.

**Questions:**

1. Could you elaborate on the advantages and disadvantages of reducing cardinality versus dimensionality?

2. Could you provide more details on how the policy network is updated without computing the Jacobian of the decoder? In Appendix B, what does $\phi$ represent in the expression $z \sim v_\phi$?

3. Could you offer some intuition regarding the trade-off among the size of the original policy network, the dimension of the latent space, and the resulting policy performance? For instance, does a larger policy network generally lead to better performance, but also require a larger encoded latent space?

---

> ### Author Response · Authors · 2025-11-21
>
> We thank the Reviewer for their detailed feedback and for recognizing the intuitive benefits of our framework. We appreciate the constructive criticism regarding the empirical metrics, which we have addressed below.
> + **W1**, **W2**. We agree that visually interpreting latent spaces is inherently subjective and that spread-based metrics alone are insufficient for a robust analytical evaluation. However, finding a scalable, quantitative metric for manifold quality remains a complex, open research challenge. We further argue that standard divergence measures would not be suitable in this context either. Our goal is not to match the input distribution, which is often biased toward low-energy random policies, but rather to remove this bias and approximate a uniform distribution over all realizable behaviors. Proving this analytically in a continuous, unbounded latent space is difficult. Therefore, we emphasize that our ultimate evaluation is the success of the latent optimization process itself. While we may lack the analytical tools to provide perfect quality estimates of the manifold (a common constraint in Deep Learning), the fact that the learned space enables fast, effective task adaptation stands as the primary evidence of its value.
> + **W3**. The observation that Latent PGPE is sometimes suboptimal compared to a *tabula rasa* baseline (like Parameter PGPE) is correct, but we view this as an intrinsic trade-off of the Unsupervised RL setting rather than a failure of expressivity. While a baseline trained from scratch acts as a “specialist” optimizing only for a specific reward, our method trains a single “generalist” latent manifold capable of solving any downstream task. The “cost” of this approach is a potential reduction in asymptotic performance compared to a specialist, but this is paid for by an increase in sample efficiency, which is our core contribution.
> + **Q1** (**Q3**).  Reducing cardinality typically transforms the task into a discrete or combinatorial optimization problem (often NP-hard), which scales poorly for both compression and downstream adaptation. In contrast, reducing dimensionality preserves a continuous, differentiable manifold. This allows us to leverage powerful gradient-based methods (like PGPE) for efficient fine-tuning. Regarding the trade-offs (**Q3**), our results indicate that the required latent dimension is dictated by the behavioral complexity of the environment, rather than the size of the policy network. Consequently, a larger policy network does not necessarily require a larger latent space. This property allows our approach to scale robustly, compressing policies with over 100,000 parameters by five orders of magnitude while maintaining functional expressivity.
> + **Q2**. In the PGPE formulation (Appendix B), the hyper policy is a distribution over the latent code $z$, denoted $\\nu_\\phi(z)$, where $\\phi$ is the vector of means and standard deviations characterizing the Gaussian distributions over the dimensions of the latent space. We sample a population of codes $z \\sim \\nu_\\phi$, decode them to parameters $\\theta = g(z)$, and evaluate them in the environment to get returns $R$. The gradient is estimated purely based on the correlation between the perturbations in $z$ and the obtained returns $R$. Since we do not backpropagate the reward signal through the decoder network $g(z)$ (we treat the environment+decoder as a black box), the Jacobian $\\nabla_z g(z)$ is not required.

---

### Official Review · Reviewer_Rt5R · 2025-10-31

**Soundness:** 2
**Presentation:** 3
**Contribution:** 2
**Rating:** 4
**Confidence:** 4

**Summary:**

The paper proposes a method of optimizing a policy not directly in parameter space, but in a lower-dimensional learned latent space. This space is learned by an autoencoder trained on a dataset of policy parameters. The training set for the autoencoder is generated using policies discovered in an 'unsupervised' (i.e., reward-free) setting via novelty search, with the L2 distance between actions on some subset of the state space serving as the distance measure. Given a reward function, a concrete policy can then be optimized directly in the learned latent space using policy gradient algorithms. The paper shows that in mountain-car-continuous and reacher environments, this latent optimization often learns faster than methods like DDPG, PPO, or SAC, which learn in parameter space.

**Strengths:**

- The proposed framework has clear intuitive benefits and is well-motivated by the manifold hypothesis.
- The experiments are clearly described and thoroughly documented.
- In the tested environments, some advantages of the method can be seen (mainly faster convergence of the policy's performance, although not always).

**Weaknesses:**

- The method is empirically tested in only two very simple continuous control settings: Mountain Car Continuous and Reacher. It is difficult to judge the promise of the method without evaluation in much more challenging and open environments.
- The method's core contribution (latent optimization) is significantly undermined if, in the best-case scenario, it simply recovers high-performing policies that were _already present_ in the training dataset generated by novelty search. The main benefit of the learned latent representation would be that it _generalizes_, i.e., that it allows for the efficient discovery of policies that are substantially different (in capability or performance) from those in the training set. It is not clearly shown in the paper that this happens in practice. While the paper mentions (lines 375-394) that some configurations recover performances _higher_ than in the training set, it is unclear if this result is statistically significant or within the bounds of evaluation noise. This point is critical to evaluating the method's true contribution.
- The paper does not mention the significant challenges and problems arising when learning latent representations of neural network parameters (e.g., scalability to large networks, or weight space symmetries). The literature on weight space learning is ignored by the paper, even though it addresses exactly this topic and might lead to significant improvements. See for example "Hyper-Representations as Generative Models: Sampling Unseen Neural Network Weights" (Schürholt et al., 2022), "Goal-conditioned generators of deep policies" (Faccio et al., 2023), or "Learning useful representations of recurrent neural network weight matrices" (Herrmann et al., 2024). Going beyond the toy domains used in the paper would likely require insights from these works.

While I find the proposed method interesting, I vote for a weak reject. I believe that a more thorough evaluation is necessary, specifically one that demonstrates statistically significant generalization beyond the training set policies and tests the method in more challenging environments.

**Questions:**

- What are the best-performing policies for the various tasks in the training datasets? A direct comparison (with confidence intervals) between the policies found by Latent PGPE and the best policies already in the training set is needed to assess the true value of the latent optimization procedure.
- Lines 359-360: "We speculate that this is due to the increased range of behaviors expressed by larger policies". Can the authors provide evidence (e.g., a quantitative measure of behavioral diversity) for an increased range of behaviors for larger policies in the training datasets?
- Lines 361-362: "...certain behavioral areas at high performance are able to grow larger, creating more nuanced decoded policies". Do larger areas in the latent space really correspond to more _behaviorally_ nuanced policies, as opposed to e.g., simply multiple different parameters encoding the same behavior (a known redundancy)? Can this be tested?
- Line 464: "bitter lesson" is obviously a loaded term in the context of RL. However, I can't see how the obvious dependency of the performance on the quality of the learned latent space relates to Sutton's essay, or why it is a "bitter lesson" in general. This seems like a straightforward and expected trade-off.
- How would the subset of the state space (used to measure policy diversity for the training set) be chosen if the state space is too large or not well-defined enough for uniform sampling?

---

> ### Author Response · Authors · 2025-11-21
>
> We thank the Reviewer for their detailed feedback and for recognizing the intuitive benefits of our framework. We particularly appreciate the push to evaluate our method in more challenging environments, which has significantly strengthened the paper.
> + **W1**. We agree that evaluating beyond simple control tasks is crucial. To address this, we have performed new experiments on the *Hopper* and *HalfCheetah* environments, which have substantially more complex observation and action spaces. The results show that our behavioral compression successfully generates rich latent spaces that can reconstruct high-performing policies across multiple tasks, even in complex environments. The results are reported in Table 2 (and below).
> + **W2**. The reviewer raises an important point about the value of the latent space. We agree that generalizing to unseen, better-performing policies would be a *nice* property; however, we don’t believe this is the only benefit of our method, or that simply recovering policies present in the dataset undermines its value. Instead, the main contribution of our latent space is to **enable** fast downstream task adaptation to any task. Recovering a high-performing policy via Latent Behavior Optimization is consistently more sample-efficient than other baselines and considerably better than searching the dataset for good policies. Still, our most recent suite of experiments provides *statistically significant evidence* that our latent space generalizes, generating policies that perform *better* than those in the training dataset. We report here (and in Table 2.) the mean ratio (and 95% CI) between the best policy sampled from the latent space and the best policy found in the dataset. We hope these results can provide confidence in our methodology’s potential to scale and generalize.
> | Environment | Task | 5D Mean | 5D 95% CI | 8D Mean | 8D 95% CI | 16D Mean | 16D 95% CI |
> | :--- | :--- | :---: | :---: | :---: | :---: | :---: | :---: |
> | **Hopper** | `forward` | 1.33 | [1.03, 1.69] | 1.59 | [1.21, 1.99] | 1.48 | [1.21, 1.78] |
> | | `backward` | 2.66 | [0.95, 5.96] | 1.29 | [1.11, 1.47] | 1.20 | [1.07, 1.33] |
> | | `standstill` | 1.49 | [1.06, 2.08] | 1.51 | [1.02, 2.25] | 1.57 | [1.07, 2.32] |
> | | `jump` | 3.83 | [0.64, 9.32] | 1.54 | [1.07, 2.14] | 2.42 | [1.35, 3.75] |
> | **HalfCheetah** | `forward` | 1.63 | [1.02, 2.31] | 1.80 | [1.20, 2.47] | 1.84 | [1.41, 2.39] |
> | | `backward` | 1.27 | [1.04, 1.55] | 1.52 | [1.19, 1.85] | 1.72 | [1.37, 2.06] |
> | | `frontflip` | 0.54 | [0.29, 0.86] | 0.74 | [0.49, 0.99] | 1.20 | [0.80, 1.63] |
> | | `backflip` | 0.55 | [0.31, 0.86] | 0.75 | [0.51, 0.99] | 1.23 | [0.81, 1.66] |
> + **W3** (**Q5**). We thank the Reviewer for highlighting this relevant literature on **Weight Space Learning** (WSL). We have expanded Appendix A (Related Works) to reflect this. We do believe that it is an important addition as there are clear parallels with our work; however, our approach sidesteps the major challenges in WSL (such as symmetries and scaling in parametrizations) by optimizing our Behavioral Loss. This naturally assimilates redundant parameterizations, similar to the concept of Policy Fingerprinting [1]. (**Q5**) Regarding the state space subset, while learning probing states is a valid future direction that incorporates insights from WSL, we suggest a simpler approach: maintaining a *Replay Buffer* for each policy and evaluating actions on the actual visited states to ensure behavioral similarity in relevant regions of the state space.
> + **Q1**. We report the Latent Optimization results against the best policies found in the original dataset in Figures 4 and 5.
> + **Q2**. We have validated this hypothesis with new empirical data in Appendix B (Figure 6). Our analysis confirms that on complex tasks (e.g., height in MC), Large policies exhibit significantly higher maximum returns (up to 800) than Small policies (up to 450). We also support our claim with relevant citations [2,3,4].
> + **Q3**. We agree that larger areas likely represent “robustness” (many codes mapping to equivalent performance). However, from an optimization perspective, larger areas in the latent space facilitate easier gradient search, which is a desirable property for the fine-tuning phase. We updated our statement accordingly.
> + **Q4**. We acknowledge that “bitter lesson” is a loaded term in RL. We intended to highlight a specific trade-off: the performance of the fine-tuning phase is strictly bounded by the quality of the unsupervised pre-training. We have rephrased the text to be more precise and avoid ambiguity.
>
> ### References
> [1] Faccio, Francesco, et al. “Goal-conditioned generators of deep policies.” AAAI, 2023.
> [2] Montùfar, Guido, et al. “On the number of linear regions of deep neural networks.” NeurIPS, 2014.
> [3] Raghu, Maithra, et al. “On the expressive power of deep neural networks.” ICML, 2017.
> [4] Bahri, Yasaman, et al. “Explaining neural scaling laws.” NAS, 2024.

---

### Official Review · Reviewer_RZJt · 2025-11-01

**Soundness:** 3
**Presentation:** 4
**Contribution:** 3
**Rating:** 8
**Confidence:** 2

**Summary:**

This paper proposes an unsupervised method to address the sample inefficiency of deep reinforcement learning. The approach learns a compact latent representation of the policy parameter space by optimizing *behavioral* similarity, allowing policies to be optimized in a low-dimensional space. Experiments on classic continuous control benchmarks demonstrate that this representation not only accelerates convergence and allows significant policy compression but also produces semantically meaningful latent representations.

**Strengths:**

* This paper studies a core problem in deep reinforcement learning: instead of optimizing the expected return over the entire space of states and actions, can we focus on visiting sttes and actions that matter for the task in hand?

* The presentation and writing quality of the paper are strong. The problem is well motivated, and the research questions are explicitly stated, making it easy to understand the focus of the study. I particularly enjoyed the background and problem formulation sections—they are cohesive, detailed, and accessible, making the paper enjoyable and understandable even for readers who may not be very familiar with the field, like myself.

* Experiments are also descent quality and help answer the main questions authors presented early in the paper.

**Weaknesses:**

* The quality of Figure 1 is quite low; the grey text is difficult to read, and zooming in causes significant pixelation, which detracts from the overall presentation of the paper.

* Figures 4 and 5 should be larger, as the curves are currently hard to distinguish. Additionally, the color choices make it difficult to differentiate the plots—for example, in Figure 4, DDPG, PPO, SAC, and TD3 all use similar green/blue colors with solid lines. Using more distinct colors and line styles would greatly improve readability.

**Questions:**

* In line 047, the term *behavior* is introduced as "a latent representation of the possible behaviors." Since its precise meaning is not clear at this point and is formally defined later in line 099, it would be helpful to emphasize it with italics to signal its special usage.

* In the experiments, it would be useful to specify the architectures used for the policies. I am particularly interested in how the depth and width vary across different policy sizes (small, medium, large), as prior work has shown that network depth can significantly affect the expressivity of deep RL policies.

* I could not identify the algorithm used for training the policies. If it is not described in the main text, this information should be added for completeness.

---

> ### Author Response · Authors · 2025-11-21
>
> We thank the Reviewer for their assessment and for appreciating the clarity, motivation, and accessibility of our work. We appreciate the Reviewer’s feedback, which has helped us improve the presentation and cohesion of the paper. We have addressed the weaknesses highlighted in the review as follows:
> + **W1**. We uploaded a higher-resolution version of Figure 1.
> + **W2**. We changed the color palette of Figures 4 and 5. We will also maximize the size of these figures in the camera-ready version within the page limit.
> + **Q1**. We implemented the suggestion to use italics for “behavior”.
> + **Q2**. We added an explicit pointer to Appendix B, where readers can find the details regarding Policy Architectures. We report here the architectures we used for:
>     - MountainCarContinuous: Small (2-4-1), Medium (2-32-32-1), and Large (2-400-300-1);
>     - Reacher: Medium policies have shape (6-64-64-2).
> + **Q3**. We have clarified which algorithm we use in the main text. To be precise, in the pretraining phase,  the policies’ weights in the Dataset are not trained but sampled from a random distribution. On the other hand, in the downstream adaptation phase, we use a modified version of Policy Gradient with Parameter-based Exploration (PGPE, [1]) that operates in the latent space rather than the parameter space.
>
> ### References:
> [1] Sehnke, Frank, et al. "Policy gradients with parameter-based exploration for control." ICANN, 2008.

---

### Meta-Review · Area_Chair_NuNi · 2026-01-09

**Summary:**

This paper proposes a technique for compressing parameters into a latent space in an unsupervised RL setting and showing this can be fused for optimizing behaviors more quickly while remaining performant. The experiments were on mountaincar and reacher.

The major concerns were:

1. Some readability/notation issues.
2. Experiments on harder tasks.
3. Only recovering policies within the training dataset.
4. ignored prior work on weight space learning.
5. Better evaluation metrics/visualizations of the latent space.

**Reviewer Concerns:**

The rebuttal mainly added some clarifications and then some experiments on halfcheetah/hopper, as well as some evidence of latent space generalization. The major concerns by the reviewers were addressed.

**Reviewer Scores:**

RZJt would remain the same at 8.
Rt5R would likely go up to a 6 given new experiments.
uJZc would likely remain the same (6) given no new experiments.

Given all of these, it tends to an accept.

---

### Decision · Program_Chairs · 2026-01-26

Accept (Poster)